# HypoBench: Towards Systematic and Principled Benchmarking for Hypothesis Generation

## Abstract

There is growing interest in hypothesis generation with large language models (LLMs). However, fundamental questions remain: what makes a good hypothesis, and how can we systematically evaluate methods for hypothesis generation? To address this, we introduce HYPOBENCH, a novel benchmark designed to evaluate LLMs and hypothesis generation methods across multiple aspects, including practical utility, generalizability, and hypothesis discovery rate. HYPOBENCH includes 7 real-world tasks and 5 synthetic tasks with 194 distinct datasets. We evaluate four state-of-the-art LLMs combined with six existing hypothesis-generation methods. Overall, our results suggest that existing methods are capable of discovering valid and novel patterns in the data. However, the results from synthetic datasets indicate that there is still significant room for improvement, as current hypothesis generation methods do not fully uncover all relevant or meaningful patterns. Specifically, in synthetic settings, as task difficulty increases, performance significantly drops, with best models and methods only recovering 38.8% of the ground-truth hypotheses. These findings highlight challenges in hypothesis generation and demonstrate that HYPOBENCH serves as a valuable resource for improving AI systems designed to assist scientific discovery.

## 1 Introduction

Hypothesis generation is ubiquitous in scientific discoveries (e.g., inferring the heliocentric model from observations of planets and moons) and in daily life (e.g., proposing reasons why one did not get admitted to college). Given the ability of LLMs to generate plausible outputs given input information, there is growing interest in exploring the promise of AI in hypothesis generation (Liu et al., 2025; Zhou et al., 2024; Ludwig & Mullainathan, 2024; Majumder et al., 2024). However, it also becomes increasingly challenging to make sense of this literature because researchers often conflate hypothesis generation with related concepts and do not have shared evaluation practice, including datasets and metrics. In this work, we aim to provide clarity on the problem of hypothesis generation and build a benchmark to enable robust progress in this emerging area.

To do that, we seek to address three key questions. First, **what is hypothesis generation?** The excitement around hypothesis generation is accompanied with the general excitement around AI for science. Therefore, it is often mixed with studies on research ideation (e.g., Si et al., 2024; Wang et al., 2024; Radensky et al., 2025). A hypothesis is a proposed explanation for a phenomenon (Wikipedia, 2025). We thus define hypothesis generation as generating natural language theories/explanations about observed phenomena (see a formal definition in § 2). This definition closely mirrors the scientific process where theories emerge from empirical observations and applies to any phenomenon that humans seek to understand, including in daily life. For instance, given observations of planets and moons, we aim to generate hypotheses (e.g., planets orbit the sun) that explain these observations. In contrast, ideation aims to generate new research directions, primarily from existing scientific literature. An example is proposing an alternative architecture to transformer. Ideation, especially in AI research, is often not about explaining a phenomenon and has a strong emphasis on differentiating from the existing literature. Recognizing this difference, our benchmark thus focuses primarily on curating observations about phenomena of interest.

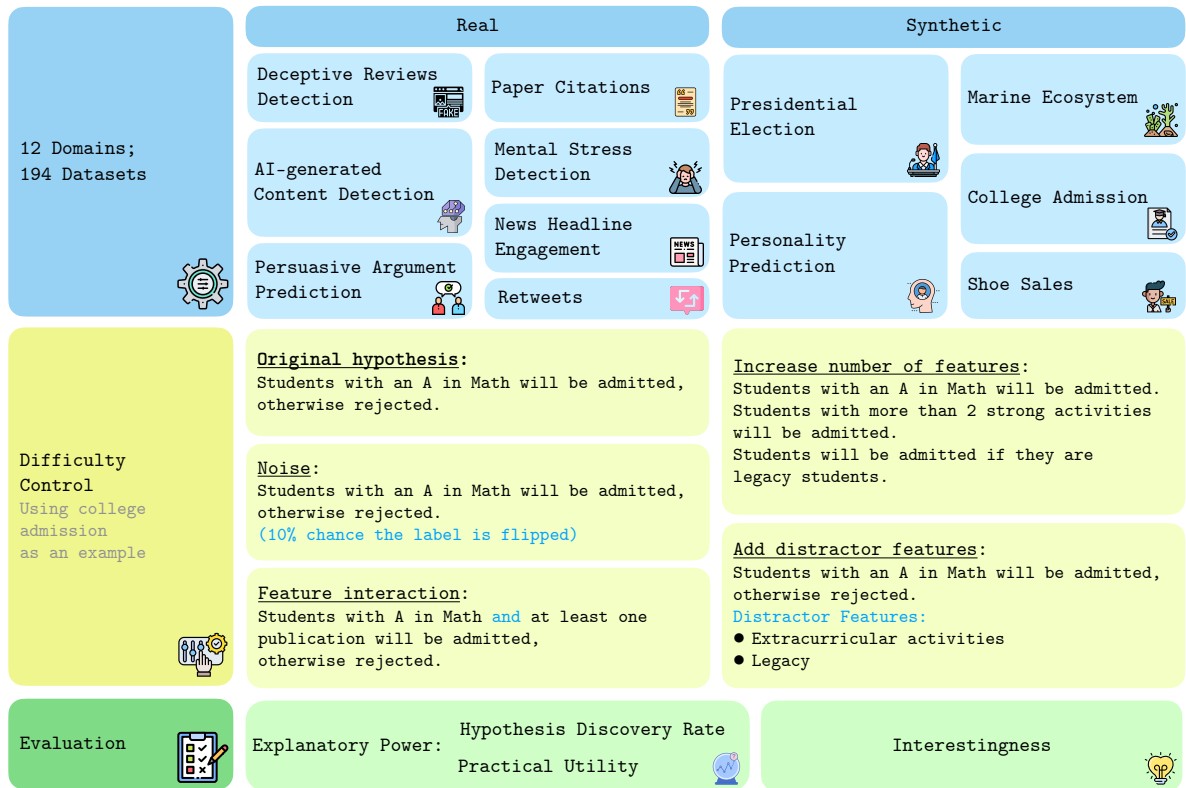

Figure 1: An overview of our benchmark. We curate 194 datasets spanning 7 real-world and 5 synthetic domains. We illustrate how difficulty levels are controlled in our synthetic settings by showing an example from the college admission task. Our evaluation measures explanatory power and interestingness of generate hypotheses.

Second, **what capabilities need to be benchmarked for hypothesis generation?** Given our focus on explaining an observed phenomenon, hypothesis generation builds on the following capabilities: 1) inductive and abductive reasoning,[1] 2) abstraction and communication, and optionally 3) synthesis, integrating new observations with existing knowledge. Inductive and abductive reasoning is necessary for proposing possible theories for a given observed phenomenon. Abstraction and communication is necessary for expressing hypotheses in natural language that humans can comprehend and appreciate. When existing literature is available, synthesis allows the models to build on relevant information. We would like our dataset to capture the complexity and diversity of hypothesis generation across different domains. In particular, we build synthetic datasets that enables arbitrary control of complexity.

Third, **how can we evaluate hypothesis generation?** Existing work (Radensky et al., 2025) tends to conflate explanatory power with novelty for hypothesis generation because it is natural for scientists to expect hypotheses to contribute to scientific advances (hence novel). However, this is not necessarily required in many other settings, e.g., proposing reasons why one did not get admitted to college. Therefore, we argue that explanatory power of hypotheses should be the first-order consideration. Interestingness and contributions to the existing literature are separate considerations, and are often subjective.[2] We focus on operationalizing explanatory power and provide preliminary measurements of "interestingness".

Building on these conceptual considerations, we build a benchmark, HYPOBENCH, by combining both real-world datasets and synthetic datasets (see Figure 1 for an overview). Our effort is highly related to DiscoveryBench (Majumder et al., 2024). The main differences are twofold: 1) DiscoveryBench assumes that relevant features have already been identified and structured, while our work captures the fact that

---

[1]HypoBench is designed to test both capabilities. Inductive reasoning involves identifying generalizable patterns from observations, while abductive reasoning involves forming higher-level theories or explanations that go beyond literal pattern discovery to explain underlying phenomena.

[2]It is well established that predicting future success is challenging (Salganik et al., 2006; Siler et al., 2015).

finding plausible features from unstructured observations is non-trivial and requires significant inductive and abductive reasoning, as well as abstraction capabilities. 2) DiscoveryBench focuses on measuring the rate of ground-truth hypothesis discovery, whereas we extend this evaluation to include additional metrics for explanatory power and preliminary metrics for interestingness.

Experiments on HYPOBENCH reveal that data-driven hypothesis generation methods outperform both zero-shot and few-shot inference across real-world and synthetic datasets. Among these methods, combining literature with data for hypothesis generation achieves the best performance. Our evaluation on real datasets shows that Qwen generates the most effective and generalizable hypotheses. However, existing methods struggle to balance plausibility and novelty in the hypotheses they generate. On synthetic datasets, the best model discovers 93.8% of the ground-truth hypotheses in base cases, but the discovery rate drops to 38.8% as difficulty increases. These findings underscore both the need for more effective hypothesis generation methods and the value of HYPOBENCH as a benchmark for advancing this line of research.

In summary, our main contributions are:

- We develop a systematic and principled framework for benchmarking hypothesis generation and construct the first benchmark accordingly.
- We conduct the first comparison between methods and models for hypothesis generation. In real-world datasets, we find that LITERATURE + DATA is the best approach and Qwen is the best model within our choices of models.
- We complement our real-world tasks with carefully controlled synthetic datasets at different complexity levels, enabling direct evaluation of how well models can recover known ground-truth hypotheses and demonstrating substantial room for improvement.

We will release all code and datasets upon publication.

## 2 Datasets and Tasks

**Problem formulation.** We start by providing a formal definition of hypothesis generation. Given a phenomenon $Q$ to understand, we assume access to a dataset $\mathcal{D}$ consisting of observations $x$ and outcomes $y$ and relevant literature $\mathcal{L}_Q$. Without loss of generality, the target variable $y$ is determined by a function $f$ on latent variables $z$, represented as $y = f(z)$. We only have access to raw observations $X$ which encode these latent variables through some mapping $g$ such that $z = g(x)$ (thus, $y = f(g(x))$). $H$, in turn, verbalizes $Z$ in ways that are interpretable to humans. The entire process of hypothesis generation can be formulated as:

$$Q, \mathcal{D}, \mathcal{L}_Q \to H,$$

where the key ingredients in building a benchmark are $Q$ and $\mathcal{D}$.

**Benchmark construction.** We use the following principles in creating the benchmark:

- The tasks and the underlying hypotheses should reflect realistic scenarios.
- The datasets should cover different skills required for hypothesis generation.
- The tasks should vary in difficulty and enable accurate evaluations of hypotheses.

Following these principles, we develop a combination of real-world and synthetic datasets. Our benchmark consists of 194 datasets spanning 12 domains—7 real-world domains and 5 synthetic domains. In particular, we curate 7 distinct real-world classification tasks by adopting datasets from prior work (deceptive reviews detection, AI-generated content detection, persuasive argument prediction, mental stress detection, news headline engagement, retweets) (Zhou et al., 2024; Liu et al., 2025) and introduces a new one (paper citations) to provide a comprehensive evaluation set (see Appendix B for details). For each task $Q$, we curate relevant literature $\mathcal{L}_Q$ that provides context for existing findings and create in-domain (IND) and out-of-domain (OOD) splits to evaluate the generalizability of discovered hypotheses. By testing on these real-world tasks, we can assess how well different methods perform on problems that reflect actual scientific inquiry challenges. While these real-world datasets provide practical validation, the true underlying hypotheses for these open problems

| Task | Description | IND Size | OOD Size |
|---|---|---|---|
| Deception Detection | Distinguish genuine and fake hotel reviews. The task requires understanding subtle linguistic cues that indicate deceptive writing. | 1,600 | 640 |
| AI-generated Content Detection | Identify whether a story is written by human or AI given a writing prompt. This tests models' ability to discover distinctive features between human and AI-generated content. | 800 | 800 |
| Persuasive Argument Prediction | Predict which text is more persuasive between pairs of arguments. The task explores linguistic features that contribute to effective persuasion in written communication. | 750 | 500 |
| Mental Stress Detection | Detecting mental stress signals from Reddit posts across different communities. This task investigates linguistic features that are indicative to mental stress in social media content. | 1,000 | 500 |
| News Headline Engagements | Given a pair of headlines of the same news article, predict which one will get more clicks from readers. | 700 | 453 |
| Retweets | Given a pair of tweets, predict which one will be retweeted more. | 1,000 | 500 |
| Paper Citations | Classify whether an academic paper will get high or low citations. | 1,182 | 1,104 |

Table 1: Overview of real-world datasets used in the hypothesis generation benchmark. IND and OOD splits are created based on different data sources or domains. Details about the IND/OOD split are provided in Table 9.

remain unknown, rendering precise evaluation challenging. This motivates our focus on the development of synthetic datasets with controlled mechanisms, which we describe in detail in the following section.

**Synthetic datasets.** Our synthetic datasets include presidential election, personality prediction, marine ecosystem, college admission, and shoe sales (see Appendix B for details). Here we provide a detailed description of how we created synthetic datasets, which enable us to precisely measure how well different methods recover true hypotheses under various controlled conditions. These synthetic datasets complement the real-world datasets by allowing systematic evaluations on ground-truth hypotheses.

As discussed above, we assume that the underlying data generating process to $f : z \to y$ is implemented via a chosen classifier and $g^{-1}: z \to x$ through prompt-driven generation. For modeling the relationship between features and outcomes, we choose logistic regression and decision trees because they are (1) interpretable building blocks widely used in scientific modeling, (2) cover both linear relationships (logistic regression) and nonlinear relationships with feature interactions (decision trees), and (3) enable explicit ground-truth hypotheses that can be precisely evaluated. Specifically, we consider:

- Logistic regression. A logistic model with $K$ classes is initialized with random weight vectors $\beta_c \in (-5, 5)$ and intercepts $\alpha_c \in (-1, 1)$ for each class $c$. The probability of class $c$ is given by $\hat{p}(y = c \mid z) = \frac{\exp(\beta_c \cdot z + \alpha_c)}{\sum_{k=1}^{K} \exp(\beta_k \cdot z + \alpha_k)}$. This approach yields a straightforward, interpretable linear decision boundary in logistic space.

- Decision tree. A small decision tree is built with randomly selected splitting features and thresholds. Each node splits on a particular feature $z_j$, with leaf nodes assigning class probabilities based on the distribution of samples that reach them. This allows for capturing nonlinear relationships and interactions among features, potentially increasing the complexity of the generated datasets.

**Abstraction layer.** Crucially, our synthetic setup is not simply "reverse-engineering a function from structured features." We introduce an abstraction layer where models observe unstructured natural-language descriptions ($x$) and must first uncover the latent variables ($z$) via abductive reasoning before discovering their relationships with the outcome ($f$). This tests the full hypothesis discovery pipeline—from raw observations to interpretable hypotheses—rather than just pattern matching on pre-structured data. This design distinguishes HYPOBENCH from benchmarks that assume relevant features are already identified and structured.

| Task | Description | Variants | Size |
|------|-------------|----------|------|
| Presidential Election | Given a person's tweet post, predict which party they will vote for the 2024 election. | 78 | 178,750 |
| Personality Prediction | Given a person's tweet, determine the personal preferences of the user based on the content, sentiment, and language patterns. | 76 | 178,750 |
| College Admission | Predicting whether a student will be admitted or not based on their background information. | 26 | 7,800 |
| Shoe Sales | Given a customer's appearance, predict the shoe they will buy. | 3 | 3300 |
| Marine Ecosystem | Given information about a marine ecosystem, predict the daily sunlight hours received per day at the location. | 1 | 500 |

Table 2: Overview of synthetic datasets used in HYPOBENCH.

**Controlling dataset difficulty.** For the synthetic datasets, we consider the following dimensions to control the difficulty of the tasks:

- Noise in the outcome. When generating the labels $Y$, some labels may be randomly flipped, or their class probabilities can be sampled to introduce randomness. This tests the models' ability to generate robust hypotheses by identifying core patterns while disregarding noise, similar to real-world scenarios.

- Number of features. Increasing or decreasing the total number of correlated features $Z$ adjusts the complexity of the dataset.

- Compositionality (depth). For decision trees, we vary the tree depth to allow for composite interactions between features.

- Distractor features. To simulate realistic settings where useful information is often mixed with irrelevant details, we consider adding distractor features $Z^0$. This evaluates the models' ability to extract truly relevant features while filtering out distracting information.

- Textual subtlety. To evaluate models' abstraction skill, we consider adding subtlety to how features are presented in the text. We create two variants: a control group with explicit feature representation and an experimental group where features are embedded within unstructured text. For example, a political feature like *endorses democratic party* might be explicitly stated in the control group, while in the subtle version it appears as *"I was quite taken aback by the criticism of the Supreme Court decision favoring conservatives, which led me to reconsider my position on championing gun rights."* This tests the models' ability to abstract and identify underlying features from natural language.

**Illustrative examples.** To help familiarize our problem formulation in addition to Figure 1, we provide concrete examples from both real-world and synthetic datasets in Table 3. We also present additional examples of generated hypotheses in Table 4. See Appendix B for more examples of input data instances.

## 3 Evaluations

To automate this process, we consider a systematic approach to evaluate hypotheses with an emphasis on explanatory power. In contrast to ideation, where novel ideas that differ from existing literature is the first-order principle, we argue that a hypothesis must first demonstrate explanatory power before its novelty becomes scientifically valuable. We evaluate explanatory power primarily through *practical utility*—whether hypotheses enable accurate predictions on held-out data—complemented by *hypothesis discovery rate* for synthetic datasets where ground-truth hypotheses are available.

**Practical utility.** Our primary measure of explanatory power is practical utility: whether discovered hypotheses support accurate prediction on held-out test data. This directly captures the "fit on unseen data" criterion central to scientific hypothesis evaluation. Specifically, given the discovered features $\hat{Z}$ and their relationships $\hat{f}$, we measure classification accuracy by prompting an LLM $\mathcal{M}_I$ to predict labels for test

**Real-World Task: Retweets**

**Observation (x):** First tweet: "CNN: Senate Democrats supported rule that led to insurance cancellations." Second tweet: "Senate Dems knew millions would receive cancellation notices, because they voted for it."
**Label (y):** 1 (second tweet got more retweets)
**Ground Truth Features (z):** Unknown
**Generated Hypothesis (h):** "Tweets that engage the audience by addressing them directly or using inclusive language tend to receive more retweets than those that focus solely on the author's perspective."

**Synthetic Task: Presidential Election**

**Observation (x):** "My belief in critiquing mainstream political parties has been strengthened by my reactions to the Supreme Court decision that I criticize for favoring conservatives. This is why I consistently express my views on opposing tax cuts for corporations using support for social justice initiatives."
**Label (y):** Democratic voter
**Ground Truth Features (z):** The text contains four key political indicators: criticizes mainstream parties (political endorsement), opposes corporate tax cuts (policy stance), supports social justice (partisan language), and criticizes conservative court decisions (political event reaction). These are encoded as a binary feature vector $z \in \{0,1\}^4$ indicating presence/absence.
**Ground Truth Predictor (f):** A multinomial logistic regression model computes the predicted label as: $f(z) = \arg\max_k (\beta_k \cdot z + \alpha_k)$, where $z \in \{0,1\}^d$ is the binary feature vector, $\beta_k \in \mathbb{R}^d$ is the weight vector for class $k \in \{$Republican, Democratic, Third-party$\}$, and $\alpha_k$ is the intercept for class $k$. In this example, features like *opposes corporate tax cuts* and *supports social justice* have positive weights in $\beta_{\text{Democratic}}$, contributing strongly to the Democratic class score and outweighing any negative contribution from *criticizes mainstream political parties*.
**Ground Truth Hypothesis (h):** "Individuals who criticize conservative institutions, advocate for social justice, and oppose corporate tax cuts are likely to support the Democratic candidate."
**Generated Hypothesis:** "Voters expressing support for Democratic policies like universal healthcare and climate action tend to indicate Democratic preference."

Table 3: Detailed examples illustrating the hypothesis generation pipeline for both real-world and synthetic datasets.

| Dataset | Hypothesis |
|---|---|
| Persuasiveness prediction (real-world) | Arguments that establish the author's expertise or authority on the subject, such as through credentials or past experience, is likely to be more persuasive. |
| Paper citations (real-world) | Abstracts that highlight novel combinations of existing knowledge or interdisciplinary approaches are more likely to indicate higher impact. |
| Presidential election (synthetic) | If a person expresses concern about climate change and the two-party system, they will more likely vote for the democratic party instead of the republican party. |
| College admission (synthetic) | If a student has at least one publication and A in Math, they will be admitted. If they are a first-generation college student, they can be admitted with B+ or higher. |

Table 4: Example hypotheses from different datasets. The ones from real-world datasets are generated hypotheses, while the ones from synthetic datasets are groundtruth hypotheses.

samples using these hypotheses:

$$\text{Accuracy}(\hat{f}, \hat{Z}, \mathbf{X}) \coloneqq \frac{\sum\limits_{(x_i, y_i) \in \mathbf{X}} \mathbb{1}(y_i = \mathcal{M}_I(x_i, \hat{f}, \hat{Z}))}{|\mathbf{X}|},$$

where $\mathcal{M}_I(x_i, \hat{f}, \hat{Z})$ represents the model's prediction when instructed to analyze input $x_i$ in terms of the discovered features $\hat{Z}$ and their relationships $\hat{f}$ with the outcome. We also compute F1 scores.

**Hypothesis discovery rate.** For synthetic datasets where ground-truth hypotheses are known, we complement practical utility with a diagnostic metric that measures how well methods recover the true hypotheses. Inspired by Majumder et al. (2024), we evaluate the hypothesis discovery rate (HDR) by combining feature discovery accuracy with relationship correctness:

$$\text{HDR} = \text{FDR} \cdot \text{RC}$$

where FDR (Feature Discovery Rate) measures the proportion of true features discovered:

$$\text{FDR} = \frac{|\hat{Z} \cap Z|}{|Z|},$$

and RC (Relationship Correctness) evaluates the accuracy of discovered relationships for the matched features:

$$\text{RC} = \frac{1}{|\hat{Z} \cap Z|} \sum_{z_i \in \hat{Z} \cap Z} \mathcal{M}_r(z_i, \hat{f}, f).$$

Here, $\mathcal{M}_r(z_i, \hat{f}, f)$ is a rating function that evaluates if the relationship between feature $z_i$ and the outcome is correctly identified in $\hat{f}$, compared to the ground-truth relationship in $f$. We employ an LLM $\mathcal{M}_r$ to rate the correctness of the discovered relationship on a scale of $[0, 1]$ by comparing the discovered relationship with the ground-truth relationship. We use GPT-4o for evaluating HDR and provide details in Appendix A. Note that HDR functions as a recall-oriented metric; we use it alongside practical utility to provide a more complete picture of hypothesis quality.

**Generalizability.** To evaluate whether generated hypotheses can generalize beyond their original context, we assess their effectiveness on data with distribution shifts. For each real dataset in HYPOBENCH, we create paired in-domain (IND) and out-of-domain (OOD) splits. We provide the details in Appendix B. We measure generalizability by computing the hypothesis-based inference accuracy and F1 scores on both the IND and OOD splits. In addition, we conduct cross-model experiments by generating hypotheses from one model and evaluating them using a different inference model. This setup tests whether hypotheses can generalize across different models. We perform these two evaluations only on the real-world datasets.

**Novelty, plausibility, and clarity** Assessing qualitative properties of hypotheses requires comprehensive understanding of scientific standards and existing knowledge. Following Liu et al. (2025), we evaluate three key properties that determine the quality of hypotheses:

- **Novelty**: The extent to which the hypothesis offers new insights beyond established knowledge in the relevant domain.
- **Plausibility**: The degree to which the hypothesis is scientifically reasonable and consistent with existing evidence.
- **Clarity**: Whether the hypothesis is clearly articulated, logically structured, and readily comprehensible.

We employ GPT-4o as a judge ($\mathcal{M}_q$) to evaluate these qualities by providing it with the generated hypotheses and relevant context from existing literature $\mathcal{L}_Q$:

$$(\text{Novelty}, \text{Plausibility}, \text{Clarity}) = \mathcal{M}_q(\hat{f}, \hat{Z}, \mathcal{L}_Q).$$

For each dimension, the model rates hypotheses on a scale of 1-5 following instructions adapted from the human expert rating study in the previous work, with detailed prompts provided in Appendix A.

To summarize, our framework addresses a key limitation in existing evaluations: the tendency to overly emphasize novelty without sufficiently assessing fundamental properties like explanatory power and plausibility.

## 4 Experiment Setup

**Models.** In this work, we evaluate four models: GPT-4o-mini (GPT), Qwen-2.5-72B-Instruct (Qwen), Llama-3.1-70B-Instruct (Llama), and DeepSeek-R1-Distilled-Llama-70B (DeepSeek). For each model, we evaluate their zero-shot and few-shot inference performance for practical utility. We also evaluate a collection of hypothesis generation methods across all metrics in § 3. In addition, due to the lack of ground-truth hypotheses, we finetune a Llama-3.1-8B model (Llama-8B) as a comparison point for each real dataset that can learn from a much larger number of instances. Specifically, for each real dataset, we finetune Llama-8B on the IND training split and evaluate it on both the IND test set and the OOD set.

**Hypothesis generation methods.** We present the first comprehensive evaluation of various hypothesis generation approaches using state-of-the-art LLMs including GPT, Llama, Qwen, and DeepSeek. We benchmark the following methods (refer to Appendix D.1 for implementation details):

| Method | GPT | | Qwen | | Llama | | DeepSeek | |
|---|---|---|---|---|---|---|---|---|
| | Accuracy | F1 | Accuracy | F1 | Accuracy | F1 | Accuracy | F1 |
| Zero-shot inference | 61.8 | 56.1 | 60.1 | 55.5 | 66.9 | 63.6 | 62.9 | 58.0 |
| Few-shot inference | 65.7 | 62.7 | 68.9 | 68.0 | 72.5 | 71.2 | 66.9 | 64.1 |
| Zero-shot generation | 62.4 | 57.6 | 63.4 | 59.1 | 62.8 | 56.4 | 62.9 | 57.8 |
| LITERATURE-ONLY | 61.9 | 57.1 | 62.5 | 57.3 | 62.0 | 55.3 | 59.3 | 53.7 |
| IO PROMPTING | 66.1 | 65.1 | 74.5 | 74.0 | 68.2 | 66.3 | 61.6 | 59.8 |
| ITERATIVE REFINEMENT | 66.0 | 63.9 | 70.5 | 69.5 | 69.9 | 68.9 | 63.6 | 62.7 |
| HYPOGENIC | 71.2 | 70.3 | 77.8 | 77.8 | 72.3 | 70.9 | 70.0 | 68.7 |
| LITERATURE + DATA | **75.3** | **75.0** | **78.0** | **77.9** | **76.2** | **75.9** | **74.9** | **74.5** |
| Finetuned Llama | OOD Accuracy: 77.3 / F1: 76.0 | | | | IND Accuracy: 84.7 / F1: 84.7 | | | |

Table 5: OOD Accuracy and F1 scores for different methods across models on real-world datasets. We report the average performance across different datasets. Standard errors across datasets are typically 2–4%, making the reported differences between methods statistically meaningful.

- Zero-shot inference. This baseline directly prompts LLMs to classify test samples based solely on the task description, without generating or using explicit hypotheses. This serves as a lower bound that tests raw model knowledge for the classification task.

- Zero-shot generation. This method first prompts LLMs to generate hypotheses from the task description alone (without seeing data examples), then uses these hypotheses for classification. Unlike zero-shot inference, this method generates explicit hypotheses that are then used to guide prediction, testing the model's ability to leverage pre-trained knowledge for hypothesis formulation.

- Literature-based generation. Several works (Wang et al., 2024; Radensky et al., 2025) have explored literature-based approaches for the ideation problem. We use the LITERATURE-ONLY adaptation from Liu et al. (2025), where the method first collects relevant research papers, prompts LLMs to summarize key findings, and then generates new hypotheses based on these insights. We evaluate this approach exclusively on the real-world datasets in our benchmark.

- IO PROMPTING (Qiu et al., 2024). This method provides a set of labeled examples from a classification task to the model and prompts it to generate hypotheses in a single step.

- ITERATIVE REFINEMENT (Qiu et al., 2024). This method builds upon IO PROMPTING by implementing a feedback loop where generated hypotheses are tested with the target classification tasks. The model uses the wrongly classified examples to refine the hypotheses, creating an iterative improvement process that enhances hypothesis quality through empirical validation. We re-implement IO PROMPTING and ITERATIVE REFINEMENT using the same hyperparameters as in the original work.

- HYPOGENIC (Zhou et al., 2024). This method implements an iterative algorithm that maintains a hypothesis bank with reward scores, balancing exploitation of high-performing hypotheses with exploration of new ones. It continuously refines the hypothesis bank by generating new hypotheses when existing ones fail on challenging examples.

- LITERATURE + DATA (Liu et al., 2025). This method extends HYPOGENIC by incorporating relevant scientific literature alongside observational data. It employs both data-analysis and literature agents during hypothesis generation, combining insights from empirical evidence and domain knowledge through iterative refinement.

# 5 Results

We present a comprehensive evaluation of hypothesis generation methods across both real-world and synthetic datasets. Our key findings are as follows: (1) Data-driven hypothesis generation methods consistently outperform simple inference approaches, with LITERATURE + DATA achieving the best performance on real datasets by effectively combining literature knowledge and empirical data. (2) Model performance

varies significantly, with Qwen excelling at hypothesis generation but showing limited ability to incorporate external literature, while Llama demonstrates strong in-context learning capabilities in few-shot settings. (3) Generated hypotheses show good cross-model generalizability, particularly between models in the same family, and achieve performance comparable to fine-tuned models on out-of-distribution data. (4) With synthetic datasets, DeepSeek achieves the best performance under base difficulty but shows high sensitivity to noise, and all models struggle with complex feature interactions beyond depth 2. (5) Model priors substantially influence hypothesis generation quality, with models performing significantly worse on counterintuitive settings.

**Evaluation results on real-world datasets (comparison between methods); Table 5.** In real-world datasets, we first observe that zero-shot generation and LITERATURE-ONLY outperforms zero-shot inference on average accuracy and F1, which suggests that the models are able to summarize useful hypothesis from its pretrained knowledge and existing literature. However, few-shot inference consistently outperforms both zero-shot generation and LITERATURE-ONLY, indicating that merely generating hypotheses based on prior knowledge is insufficient for generating effective hypotheses.

Additionally, we observe that data-driven hypothesis generation methods including IO PROMPTING, ITERATIVE REFINEMENT, HYPOGENIC, and LITERATURE + DATA all outperform few-shot inference. This improvement demonstrates that these approaches to hypothesis generation are capable of extracting and synthesizing more useful information from the data than a few examples alone. Among these methods, LITERATURE + DATA achieves the best performance, highlighting the complementary benefits of integrating both literature knowledge and empirical data for hypothesis generation.

**Evaluation results on real-world datasets (comparison between models); Table 5.** When it comes to models, they perform differently in a few ways: Llama achieves the best performance when using few-shot inference, outperforming the other models by 5.3% on average accuracy, demonstrating strong in-context learning capabilities. Interestingly, Qwen achieves the best performance when using the strongest hypothesis generation method (LITERATURE + DATA), surpassing the other models by an average accuracy of 2.5%. This suggest that Qwen is particularly good at coming up with effective and generalizable hypotheses.

Notably, Qwen shows an interesting trend: it gains little from the inclusion of literature information. While the average improvement across all other models when adding literature to data-driven hypotheses is 4.3%, Qwen improves by only 0.1%. This highlights a potential limitation in Qwen's ability to incorporate external knowledge during hypothesis generation.

| Gen \ Inf | GPT | Qwen | Llama | DeepSeek |
|---|---|---|---|---|
| GPT | 75.3 | 69.0 | 64.5 | 67.9 |
| Qwen | 64.7 | 78.0 | 68.3 | 74.4 |
| Llama | 66.1 | 74.8 | 76.2 | 72.6 |
| DeepSeek | 65.7 | 75.0 | 72.4 | 74.9 |

Table 6: Cross-model hypothesis-based inference accuracy for OOD data. Row indicates the model used to generate the hypotheses, while Column indicates the model used to infer the outcome from the generated hypothesis.

Table 6 shows the cross-model inference performance. Hypotheses generated by Qwen, Llama, and DeepSeek generalize well across models within this subgroup, with an average accuracy drop of only 3.4% compared to using the original generation model. This pattern is particularly strong between Llama and DeepSeek, likely because DeepSeek-R1-Distilled-Llama belongs to the same model family as Llama. In contrast, GPT seems to differ substantially from the other models.

When comparing the performance of hypothesis generation with fine-tuned Llama, all hypothesis generation methods perform on par with the finetuned Llama-8B on the OOD datasets, sometimes even marginally better, with Qwen leading by 0.6%. This observation further validates the effectiveness of current hypothesis generation approaches. However, the number is substantially lower in IND, where the best model Qwen underperforms by 8.7% (see the full IND results in Table 12). Since we do not know the groundtruth hypotheses in these real-world datasets, this gap could suggest potential room for further improvements as the fine-tuned Llama is capable of achieving higher IND performance. This inconclusiveness about upbound further motivates our creation of synthetic datasets.

**Qualitative ratings of hypotheses in real-world datasets.** Table 7 shows that on average, LITERATURE-ONLY generated hypotheses score highest in terms of plausibility but lowest in novelty. This may because the models are likely pretrained on similar data and thus generate hypotheses that are similar to existing

| Method | GPT | | | Qwen | | | Llama | | | DeepSeek | | |
|---|---|---|---|---|---|---|---|---|---|---|---|---|
| | N | P | C | N | P | C | N | P | C | N | P | C |
| Zero-shot generation | 2.41 | 4.04 | 3.29 | 2.11 | **4.21** | 3.31 | 2.59 | 4.01 | 3.36 | 2.27 | 4.06 | **3.27** |
| LITERATURE-ONLY | 2.11 | 4.14 | 3.44 | 2.00 | 4.20 | 3.60 | 2.21 | **4.14** | 3.37 | 1.96 | **4.20** | 3.20 |
| IO PROMPTING | 2.54 | 3.77 | 3.43 | 2.63 | 3.77 | 3.34 | 2.57 | 4.06 | 3.46 | 2.46 | 3.77 | 3.26 |
| ITERATIVE REFINEMENT | **2.97** | 3.83 | 3.14 | **2.86** | 3.49 | 3.11 | **2.74** | 3.83 | 3.37 | **2.63** | 3.86 | 3.14 |
| HYPOGENIC | 2.68 | 3.98 | 3.39 | 2.46 | 3.94 | 3.43 | 2.46 | 3.94 | 3.29 | 2.51 | 3.91 | 3.10 |
| LITERATURE + DATA | 2.69 | **4.22** | **3.58** | 2.41 | 4.14 | **3.66** | 2.58 | 4.10 | **3.52** | 2.23 | 4.02 | 3.08 |

Table 7: Qualitative evaluation results for different methods across models. We report the novelty (N), plausibility (P), and clarity (C) ratings of the generated hypotheses.

knowledge. In contrast, ITERATIVE REFINEMENT achieves the highest novelty, potentially due to its iterative refinement process that encourages the model to generate hypotheses that are distinct from the initial ones. Overall, we see that balancing plausibility and novelty is a challenging task for hypothesis generation methods, and there is no single method that excels in both metrics.

**Evaluation results on synthetic datasets.** Synthetic datasets enable ground-truth evaluation that is impossible with real-world data. While performance on synthetic tasks does not directly predict real-world discovery capability, these controlled experiments diagnose specific model limitations: sensitivity to noise, handling of feature interactions, robustness to distractors. Those aspects are relevant to real discovery but cannot be precisely measured without ground truth.

Given that synthetic datasets have no existing literature by design, we evaluate only the data-driven hypothesis generation methods. In this evaluation, we focus primarily on HYPOGENIC, the best-performing data-driven method, and analyze its performance across four different models. Figure 2 reveals clear trends in model performance as task complexity increases. With tasks at base difficulty level, i.e., one groundtruth feature, depth-1, and no noise or distractors, DeepSeek achieves the best performance among all four models, having a near-perfect HDR score of 93.8% and effectively capturing most ground-truth hypotheses. In contrast, GPT gets the lowest HDR score of 75.0% in the base difficulty.

However, we observe a significant drop in DeepSeek's performance with increased noise in outcomes (Figure 2c) and additional distractor features (Figure 2d), with HDR dropping to 40.0% and 38.3%, respectively. Interestingly, this performance drop is larger for DeepSeek compared to the other models, suggesting that its internal reasoning or "thinking mode" is particularly sensitive to noisy conditions. GPT, on the other hand, gets affected by noise in outcome and distractors slightly less, achieving HDR scores of 36.2% and 41.7%, respectively. Combined with GPT's base difficulty performance, this result may suggest that GPT generates less diverse hypotheses, hence not fully capturing all hypotheses in base difficulty but slightly more robust under the effect of noise.

Additionally, we investigate the impact of compositionality (i.e., feature interaction) in ground-truth hypotheses (Figure 2b). We see that increasing the complexity from depth 1 to depth 2 does not substantially impact model performance. Instead, Qwen and Llama are able to achieve much higher performance in depth 2 compared to depth 1, with improving HDR scores from 81.3% to 93.8%, and 87.5% to 100%, respectively. This suggests that Qwen and Llama are more likely to capture interactions between two features. However, a further increase from depth 2 to depth 3 and 4 significantly reduces HDR scores for all models, and the best model DeepSeek, in this configuration, only achieves HDR score of 38.8%. This indicates that current models can effectively discover hypotheses involving interactions between two features but face substantial challenges against more complex feature interactions. In Figure 2e, we compare the average HDR scores across all tasks without subtlety versus with subtlety in the input texts. The performance drop for all four models highlights the additional difficulty of hypothesis generation when the underlying features are implicit.

In Figure 3, we compare the performance of four models on four synthetic tasks: PRESIDENTIAL ELECTION, PERSONALITY PREDICTION, COLLEGE ADMISSION, and SHOE SALES (see dataset details in Appendix B). For each task, we present aggregated results for both the base difficulty and hardest difficulty datasets. We observe that in zero-shot generation, none of the models effectively recover the ground-truth hypotheses (HDR score < 20%) for the PRESIDENTIAL ELECTION (Figure 3e) and PERSONALITY PREDICTION (Figure 3f). Conversely,

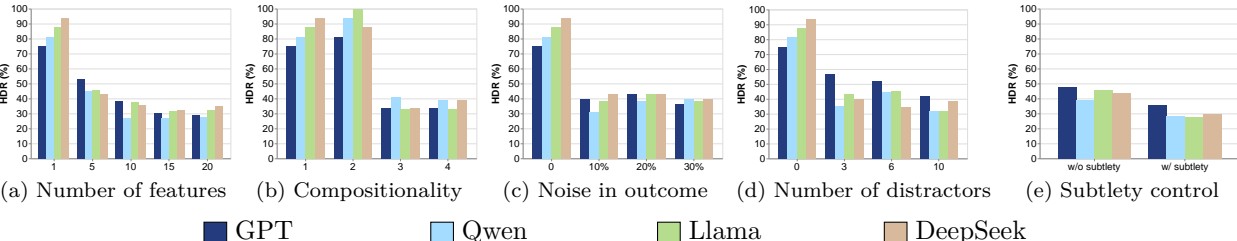

Figure 2: HypoGeniC hypothesis discovery rate (HDR) results on synthetic datasets with different task difficulty. As task difficulty increases, HDR substantially drops, even to below 30% sometimes.

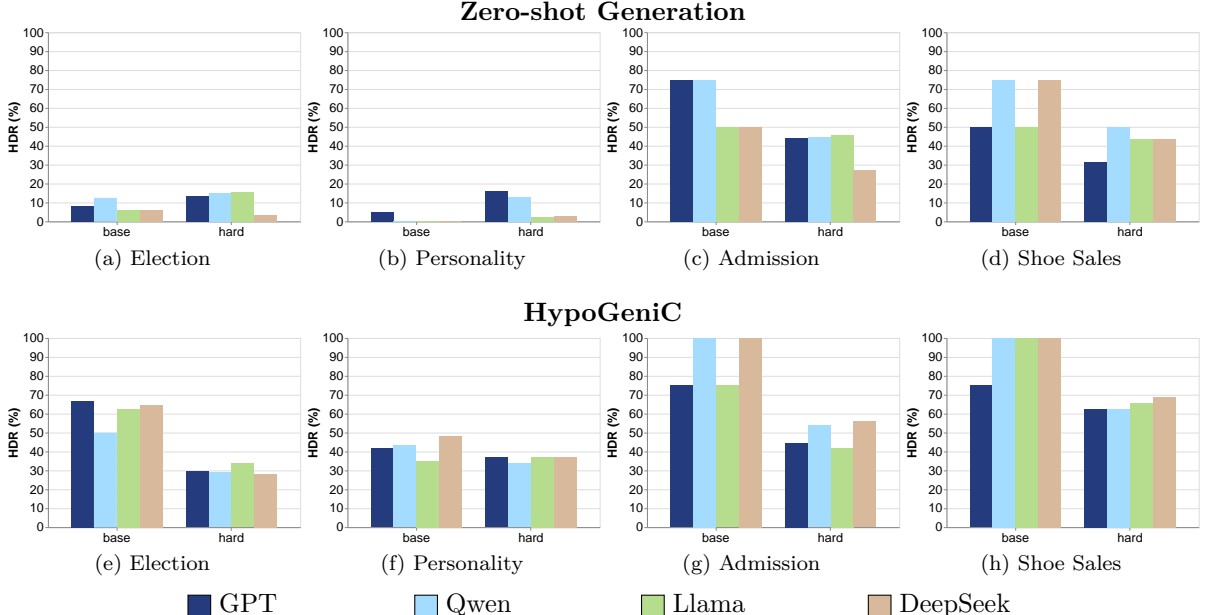

Figure 3: HDR scores of Zero-shot Generation and HypoGeniC on four different synthetic datasets: Presidential Election, Personality Prediction, College Admission, and Shoe Sales. The results show that model priors can affect the quality of the generated hypotheses in different datasets.

all models successfully discover some ground-truth hypotheses in College admission and Shoe Sales tasks, achieving HDR scores exceeding 50% at base difficulty. This suggests that the models' prior knowledge aligns more closely with College admission and Shoe Sales tasks than with Presidential election and Personality prediction. This trend is consistent with HypoGeniC, where Llama and DeepSeek achieve perfect HDR scores (100%) at the base difficulty of the College admission task (Figure 3g), and Qwen, Llama, and DeepSeek similarly achieve HDR scores of 100% for the Shoe Sales task (Figure 3h). In contrast, the models perform relatively poorly on Presidential election and Personality prediction. Specifically, GPT and DeepSeek, the best-performing models at base difficulty, achieve HDR scores of only 66.7% and 48.3%, respectively. For the harder levels, all models yield HDR scores below 40% across both tasks. These findings underscore the significant influence of model priors on hypothesis generation quality, emphasizing the utility of HypoBench's diverse task settings for evaluating different model priors.

To further explore the impact of model priors, we conduct an additional experiment comparing original and counterintuitive versions. Specifically, for the College admission task, we create counterintuitive counterparts at all difficulty levels by inverting ground-truth hypotheses (e.g., from "Students with an A in Math will be admitted" to "Students with an F in Math will be admitted"). In Figure 4, we report HDR scores for HypoGeniC across all models and levels for both the original and counterintuitive datasets.

We find that when the number of features and compositionality are low (1 or 5 features, depth 1 or 2), all models except GPT still manage to capture the ground-truth hypotheses (Figures 4a, 4b, 4e and 4f). However, as complexity increases, all four models struggle significantly, achieving average HDR scores below

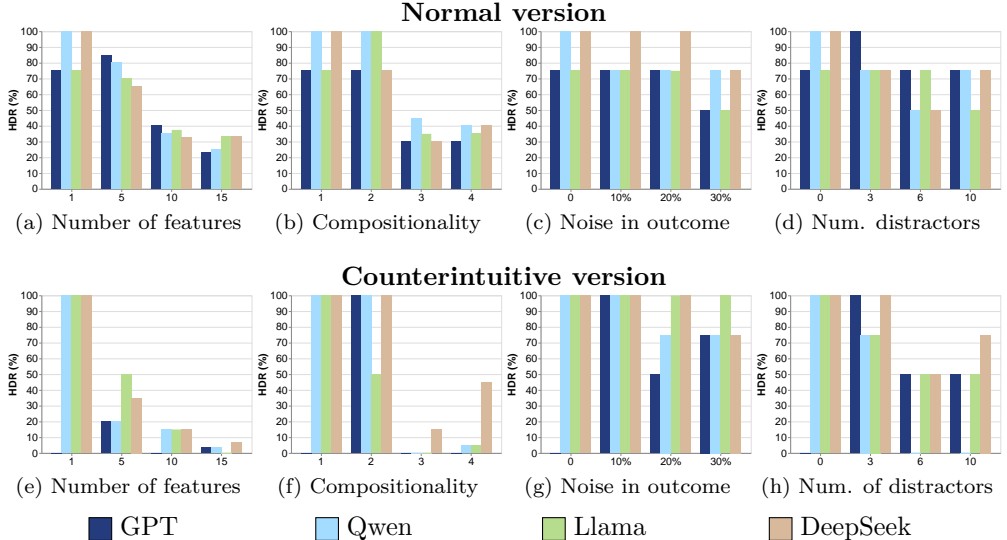

Figure 4: HYPOGENIC HDR scores on the College Admission datasets under different difficulty controlls. Top: normal ground-truth hypotheses; bottom: counterintuitive ground-truth hypotheses.

15%. Notably, DeepSeek consistently outperforms the other models in counterintuitive scenarios with greater complexity, suggesting that DeepSeek's "thinking-mode" enhances performance when model priors provide little guidance. Additionally, increasing noise in outcome and number of distractor variables primarily impact Qwen, leading to HDR scores dropping to 0% under high distractor conditions (Figures 4c, 4d, 4g and 4h). Overall, these results highlight the effect model priors have on hypothesis generation quality, particularly under challenging conditions where priors are less helpful. This further demonstrates HYPOBENCH's value as a comprehensive benchmark for evaluating diverse models and methods.

**Validating LLM-based evaluation.** To validate the reliability of our LLM-based HDR metric, we conducted a human annotation study comparing human judgments against LLM-as-judge scores. We sampled 100 hypothesis pairs from evaluation logs using stratified sampling across task categories and score ranges. Two annotators independently assessed Feature Discovery Rate (binary) and Relationship Correctness (5-point scale), with LLM scores hidden to prevent bias. Table 8 shows the results. Human-human agreement was substantial to almost perfect ($\kappa = 0.80$ for FDR, weighted $\kappa_w = 0.86$ for RC), indicating well-defined annotation criteria. Model-human agreement was also substantial ($\kappa = 0.71$ for FDR, $\kappa_w = 0.64$ for RC), with 78% of RC predictions falling within one point of human consensus. These results confirm that LLM-based evaluation provides reliable approximations of human judgment.

| Metric | Measure | Human-Human | Model-Human |
|---|---|---|---|
| FDR | Cohen's $\kappa$ | 0.80 | 0.71 |
| | % Agreement | 92.0% | 89.1% |
| RC | Weighted $\kappa$ | 0.86 | 0.64 |
| | Within 1-point | 97.1% | 78.1% |

Table 8: Inter-annotator agreement for HDR evaluation. Human-human agreement is substantial to almost perfect; model-human agreement is substantial, validating LLM-as-judge reliability.

# 6 Related Work

**Benchmarks for research tasks.** As there are growing interest in leveraging LLMs in scientific research, various benchmarks emerged for evaluating LLMs' capability in research tasks. These include agentic frameworks for data analysis (Majumder et al., 2024; Gu et al., 2024; Hu et al., 2024; Chen et al., 2024; Huang et al., 2024; Guo et al., 2024b), literature processing and information retrieval (Press et al., 2024;

Ajith et al., 2024; Kang & Xiong, 2024; Zhang et al., 2024), and broader scientific research tasks (Tian et al., 2024; Jansen et al., 2024).

In addition to DiscoveryBench (Majumder et al., 2024), other related benchmarks including Guo et al. (2024a), which benchmarks LLMs' ability to generate the core ideas of a target paper by providing the source literature that inspired it. Jansen et al. (2024) assesses LLM-driven scientific discovery pipelines in fully synthetic environments, and Hua et al. (2025) evaluates LLMs in synthetic inductive reasoning tasks. Additionally, Zhong et al. (2023) introduces D5, a task for goal-driven discovery of distributional differences between text corpora. D5 produces natural language descriptions of how two datasets differ (e.g., "patients taking drug A mention paranoia more often") and evaluates validity and goal-relevance. While related, D5 focuses on comparative corpus analysis rather than generating explanatory hypotheses for observed outcomes. Our work complements this by emphasizing explanatory power via practical utility and HDR, and by testing abstraction capabilities where latent features must be inferred from unstructured observations.

**Research agents.** Aside from the benchmarks, there are numerous recent work that aim to build LLM-powered agents to assist scientific research. Baek et al. (2024), Wang et al. (2024), and Radensky et al. (2025) focus on the ideation problem and propose methods for generating novel research ideas from existing literature, while Zhou et al. (2024) and Liu et al. (2025) introduce frameworks for generating hypotheses to explain real-world phenomena. In addition, some recent work explore automating the complete research process using LLMs (Lu et al., 2024; Li et al., 2024).

**Additional hypothesis generation methods.** Concept Bottleneck Models (CBMs) achieve explainability by routing predictions through intermediate, human-readable features (Koh et al., 2020). Recent advancements generalize this by moving beyond manually crafted features. Zhong et al. (2024) employ statistically learned natural-language predicates, discretized via LLM prompts, for diverse tasks such as clustering and classification. Dunlap et al. (2024) generate candidate sentences using GPT-3, optimizing for diversity and discriminability, and align them with images using CLIP. Schrodi et al. (2024) propose unsupervised sparse activation bases (UCBM) for enhanced interpretability. Movva et al. (2025) introduce HYPOTHESAES, a method combining sparse autoencoders (SAEs) trained on text embeddings with an LLM to produce interpretable hypotheses relating text to a target variable. The key innovation arises from using the SAE as a selective filter, reducing the entropy of input data before LLM processing.

Despite these advances, current evaluation approaches largely rely on human judgment or LLM-as-a-judge, and the question of what constitutes a good hypothesis still remains. We introduce HYPOBENCH as a standardized benchmark for evaluating LLMs and hypothesis generation methods across multiple dimensions, aiming to support more rigorous and principled development in this space.

# 7 Conclusion

In this work, we present HYPOBENCH, a principled benchmark for evaluating hypothesis generation methods across real-world and synthetic tasks. HYPOBENCH offers the first systematic evaluation of what makes a good hypothesis by assessing multiple dimensions such as explanatory power, practical utility, and generalizability. Our results show that while existing methods provide some explanatory value and outperform few-shot inference, there remains substantial room for improvement. These findings underscore the need for more effective hypothesis generation approaches and position HYPOBENCH as a valuable resource for future research. For future work, we consider extending the types of tasks and dataset structures in HYPOBENCH to include broader and more general observations, such as scientific reports and physical environment observations, thereby enhancing its utility for diverse scientific discoveries.

# 8 Limitations

**Evaluation methodology.** Our evaluation relies on LLM-as-a-judge approaches for both the HDR metric and qualitative ratings (novelty, plausibility, clarity). We designed HDR to decompose evaluation into simpler sub-tasks (feature matching and relationship correctness), and a human annotation study shows substantial model-human agreement ($\kappa = 0.71$ for FDR, weighted $\kappa_w = 0.64$ for RC). For the qualitative metrics, we did

not conduct human validation, and these ratings should be interpreted as preliminary signals rather than definitive assessments. To ensure robust evaluation, we emphasize practical utility as our primary metric, which directly measures predictive performance on held-out data without relying on LLM judgments.

**Scope of evaluation.** HDR measures whether ground-truth hypotheses are recovered, functioning as a recall-oriented metric rather than a comprehensive measure of explanatory power. We complement HDR with practical utility to assess whether hypotheses support accurate prediction. Additionally, our benchmark focuses on binary classification settings; extending to regression or open-ended scientific discovery remains future work.

**Synthetic versus real-world tasks.** Our synthetic datasets are designed to provide controlled, ground-truth evaluation rather than to directly predict performance on real-world discovery tasks. The gap between synthetic and real performance should be interpreted as a diagnostic signal about model capabilities under controlled conditions, not as a claim about transfer to real scientific settings. We note that even O3, a state-of-the-art reasoning model, achieves only 0.52 average HDR on a subset of synthetic datasets (Appendix C.4), suggesting that HYPOBENCH remains a challenging benchmark for continued progress.

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

# A  Prompts

All our prompts for LLMs are separated into system prompts and user prompts. System prompts contain role and tone information, followed by detailed descriptions of the task and the expected response format. User prompts contain useful information for downstream tasks such as dataset generation or hypothesis evaluation.

## A.1  Synthetic Dataset Generation

System Prompt
You are an expert in classification tasks and synthetic dataset creation for social science research. Your task is to generate a diverse and meaningful set of classification labels for a given task. The labels must be:
1. Relevant to the task and grounded in the provided context.
2. Diverse enough to cover typical, nuanced, and counterintuitive classifications.
3. Actionable for further data generation steps, ensuring they capture the complexity of social science phenomena.

User Prompt
Here is the description of the classification task:

<task_description>

Your task is to generate <num_labels> classification labels relevant to this task. Ensure the labels:
1. Are specific and meaningful for the task.
2. Capture diverse aspects, including typical and counterintuitive classifications, where applicable.
3. Are clearly described to ensure they can guide the generation of features and synthetic text.

Please list the classification labels and provide a brief explanation for each.

Synthetic Dataset Label Generation

System Prompt
You are a social scientist tasked with creating a synthetic dataset for the task that will be provided by the user.
To do this, you need to define a set of phrase types that can be used as placeholders in generated textual templates. These phrase types should be relevant to the task's expression format and the provided prediction labels.

Example: As a person with [gender], I advocate [advocation] and support [opinion], where "gender", "advocation", and "opinion" are phrase types, acting as placeholders in the generated textual templates.

User Prompt
Here is the description of the classification task and the specified labels:
<task_description>

Labels:
<labels>

Your task is to define <num_blanks> phrase types that can be used as placeholders in the generated textual templates following the instructions provided in the system prompt.
Please list the phrase types that are relevant to the provided task and labels.

Synthetic Dataset Template Blank Type Generation

System Prompt
You are an expert in feature engineering for social science research and synthetic dataset creation. Your task is to generate a diverse set of descriptive phrases for a given classification task with labels and a phrase type as a placeholder given from a text template.
The generated phrases must:
1. Be relevant to the task and grounded in social science principles.
2. Be diverse and representative of the classification labels and the phrase type provided.
3. Be described in a way that facilitates the generation of synthetic text and classification.

Example: if the phrase type is "opinion" and one of the labels is "Democrat", one possible generated phrase could be "supports progressive policies".

User Prompt
Here is the description of the classification task and the specified labels:
<task_description>

Labels:
<labels>

Phrase Type:
<phrase_type>

Generate <num_phrases> descriptive phrases relevant to this task following the instructions provided in the system prompt.

Synthetic Dataset Feature Generation. Here each feature is represented by a phrase.

---

System Prompt
You are a social scientist tasked with creating a synthetic dataset for the task that will be provided by the user.
To do this, you need to generate a set of textual templates that can be used to create synthetic text data. These templates should be relevant to the task's expression format and the provided prediction labels.
Each template should leave blankspaces for filling the set of phrase types provided by the user.
Example: if the user provided phrase type "gender", the prompt template generated must have a blank space available for filling this information.
E.g, the generated template could include components like: "As a person with [gender], ...", where "[gender]" is a placeholder for the phrase type.
You must mark all the blankspaces in the template with the phrase type provided by the user, as well as square brackets to indicate the placeholder, like the example shown above.

User Prompt
Here is the description of the classification task and the specified labels:
<task_description>

Labels:
<labels>

The data template must leave blankspaces for filling the set of phrase types shown below:
<phrase_types>

Generate <num_templates> textual templates that can be used to create synthetic text data following the instructions provided in the system prompt. All templates must include placeholders for ALL the provided phrase types.
Remember to mark all the blankspaces in the template with the phrase type provided by the user and square brackets to indicate the placeholder.

---

Synthetic Dataset Templates Generation

---

System Prompt
You are an expert in social science research and synthetic dataset creation. Your task is to enhance the grammar of a given phrase for a specific phrase type in a generated text prompt template.

You will perform the operations following a series of steps.
Step 1: remind yourself the original text template by reiterating it here, along with the explicit instruction "not to modify any single letter of the template".
Step 2: rewrite those phrases in the sentences with enhanced grammar for the specified phrase type, without changing the template.
Step 3: list those modified phrases from the sentences.

User Prompt
You are an expert in social science research and synthetic dataset creation. Your task is to enhance the grammar of a given phrase for a specific phrase type in a generated text prompt template.

Here is the description of the classification task, the text template, and user specified phrase-type and corresponding phrases:
<task_description>

Text Template:
<text_template>
Phrase Type: <phrase_type>
Phrases:
<phrases>

Direct plug in sentences:
<rewritten_texts>

You should output your responses step by step following the instructions below:
First, remind yourself with the text template and the instruction to never replace any components of it, including all the square brackets for parsing by explicitly stating it in your response.
Then rewrite the above direct plug in sentences with only modifications for the phrases within the blankspace "[<phrase_type>]" to ensure grammatical coherence with the surrounding text. The modified phrases should take the form "[modified phrase]", where the square brackets are explicitly kept. You may consider using clauses.
Your rewritten sentences must keep the square brackets of "<phrase_type>", instead of removing it and erroneously changing the surrounding text of the template for this phrase_type.

---

Synthetic Dataset Feature Grammar Enhancement

## A.2 Hypothesis Evaluations

**Qualitative evaluations of the generated hypotheses in real datasets.** For evaluating the generated hypotheses in terms of novelty, plausibility, and clarity for the real datasets, we follow Liu et al. (2025)'s study and adopt their instructions given to human experts. We show the exact prompts below.

---

System Prompt
You are an expert evaluator analyzing the novelty of scientific hypotheses.
Your task is to evaluate how novel the hypothesis is compared to existing knowledge.

Novelty Scale (1-5):
1: Not novel - The hypothesis has already been shown, proven, or is widely known, closely mirroring existing ideas without introducing any new perspectives.

2: Minimally novel - The hypothesis shows slight novelty, introducing minor variations or nuances that build upon known ideas but do not offer significant new insights.

3: Moderately novel - The hypothesis demonstrates moderate novelty, presenting some new perspectives or angles that provide meaningful, but not groundbreaking, avenues for exploration.

4: Notably novel - The hypothesis is notably novel, offering unique nuances or perspectives that are well-differentiated from existing ideas, representing valuable and fresh contributions to the field.

5: Highly novel - The hypothesis is highly novel, introducing a pioneering perspective or idea that has not been previously explored, opening entirely new directions for future research.

*User Prompt*
Evaluate the novelty of this hypothesis compared to existing knowledge:

Existing Knowledge:
<known_hypotheses>

Hypothesis: <hypothesis>

Format your response as:
Score: [1-5]
Reasoning: [explanation]

Scientific Hypothesis Evaluation: Novelty

---

*System Prompt*
You are an expert evaluator analyzing the plausibility of scientific hypotheses.
Your task is to evaluate if the hypothesis makes logical sense and aligns with scientific reasoning.

Plausibility Scale (1-5):
1: Not plausible - The hypothesis does not make sense at all, lacking logical or empirical grounding and failing to align with established knowledge or principles.

2: Minimally plausible - The hypothesis has significant plausibility challenges, making sense in limited contexts but contradicting existing evidence or lacking coherence with established theories.

3: Moderately plausible - The hypothesis makes sense overall and aligns with general principles or existing knowledge but has notable gaps or uncertainties that raise questions about its validity.

4: Mostly plausible - The hypothesis is mostly plausible, grounded in logical reasoning and existing evidence, with only minor uncertainties or assumptions that could reasonably be addressed.

5: Highly plausible - The hypothesis is highly plausible, fully aligning with established knowledge and logical reasoning, will likely be supported in experiments or theoretical consistency, and highly likely to be true.

*User Prompt*
Evaluate the plausibility of this hypothesis:

Existing Knowledge:
<known_hypotheses>

Hypothesis: <hypothesis>

Consider:
- Does it make logical sense?
- Are the relationships reasonable and consistent with known patterns?
- Does it align with or reasonably extend existing knowledge?
- Could this be tested?

Format your response as:
Score: [1-5]
Reasoning: [explanation]

Scientific Hypothesis Evaluation: Plausibility

---

*System Prompt*
You are an expert evaluator analyzing the clarity of scientific hypotheses.
Your task is to evaluate how clearly and unambiguously the hypothesis is stated.

Clarity Scale (1-5):
1: Highly ambiguous - The hypothesis is presented in a highly ambiguous manner, lacking clear definition and leaving significant room for interpretation or confusion.

2: Somewhat clear but vague – The hypothesis is somewhat defined but suffers from vague terms and insufficient detail, making it challenging to grasp its meaning or how it could be tested.

3: Moderately clear – The hypothesis is stated in a straightforward manner, but lacks the depth or specificity needed to fully convey its nuances, assumptions, or boundaries.

4: Clear and precise – The hypothesis is clearly articulated with precise terminology and sufficient detail, providing a solid understanding of its assumptions and boundaries with minimal ambiguity.

5: Exceptionally clear – The hypothesis is exceptionally clear, concise, and specific, with every term and aspect well-defined, leaving no room for misinterpretation and fully encapsulating its assumptions, scope, and testability.

User Prompt
Evaluate the clarity of this hypothesis in the context of existing knowledge:

Existing Knowledge:
<known_hypotheses>

Hypothesis: <hypothesis>

Consider:
- Are all terms and concepts precisely defined?
- Is the relationship between variables explicitly stated?
- Is there any ambiguity that could lead to multiple interpretations?

Format your response as:
Score: [1-5]
Reasoning: [explanation]

---

Scientific Hypothesis Evaluation: Clarity

**Hypothesis Discovery Rate Evaluation Prompts.** Here we provide the prompts we use to evaluate hypothesis discovery rate (HDR) for the synthetic datasets. We separate the feature discovery rate (FDR) and relationship correctness (RC) in the following:

System Prompt
You are an expert evaluator analyzing hypotheses about relationships between input variables (features) and predicted outcomes (labels/classes).

Your task is to identify when two hypotheses discuss the SAME INPUT VARIABLE(S) or FEATURE(S).

Important instructions:
- Match ONLY based on input variables/features discussed.
- DO NOT match based on predicted outcomes, labels, or classes.
- For hypotheses mentioning multiple variables/features, respond 'yes' if ANY input variable matches.
- Ignore the direction, thresholds, or specific values of the relationships.
- Predicted outcomes or labels must be completely ignored when determining matches.
- Empty or invalid hypotheses should always return 'no'.

Examples:

Hypothesis A: 'Students with high math scores and 2+ publications are admitted.'
Hypothesis B: 'Students with high math scores are rejected.'
Return: 'yes' (matching input variable: math scores)

Hypothesis A: 'Users who frequently watch science documentaries tend to be classified as science enthusiasts.'
Hypothesis B: 'Users mentioning climate change tend to be classified as science enthusiasts.'
Return: 'no' (the first discusses 'watching documentaries', the second discusses 'mentioning climate change'; the matching label 'science enthusiasts' is irrelevant)

Hypothesis A: 'If entertainment preference is watching health-related TV shows, users are classified as Health-Conscious Eater.'
Hypothesis B: 'Expressing enthusiasm for outdoor activities indicates health-conscious eating.'
Return: 'no' (entertainment preference vs. outdoor activities; shared labels like Health-Conscious Eater should NOT count as matching)

Responses must be exactly 'yes' or 'no'.

User Prompt
Determine if these two hypotheses discuss any of the same INPUT VARIABLES or FEATURES.

True Hypothesis: <hyp_true>
Generated Hypothesis: <hyp_gen>

Remember:
- DO NOT consider predicted outcomes, labels, or classes.
- Focus ONLY on the input variables/features being discussed.
- Ignore relationship directions, thresholds, or specific values.
- Respond 'yes' if ANY input variable is shared; otherwise, respond 'no'.

- Return 'no' for empty or invalid hypotheses.

Response should be exactly 'yes' or 'no'.

---

Hypothesis Evaluation: Feature Discovery Rate

---

System Prompt

You are an expert evaluator analyzing hypotheses about relationships between input variables (features) and predicted outcomes (labels/classes).

Your task is to evaluate how correctly a generated hypothesis captures the relationships described by the true hypothesis.

Important guidelines:
- Evaluate BOTH the variables/features AND the direction or nature of their relationships to predicted outcomes.
- Clearly contradictory relationships should always receive a score of 0.0.
- For composite hypotheses (multiple conditions), assign partial scores proportionally.
- Ignore irrelevant additional information if the main relationships and conditions are accurately captured.
- Empty or invalid hypotheses always score 0.0.

Scoring Examples:

True: 'Students with A in math AND 2+ publications are admitted.'
Generated: 'Students with A in math are admitted.'
Score: 0.5 (captures one of two conditions)

True: 'Students with A in math will be admitted.'
Generated: 'Students with F in math will be admitted.'
Score: 0.0 (clearly contradictory relationship)

True: 'Users watching health-related shows prefer health-conscious eating.'
Generated: 'Users who enjoy hiking prefer health-conscious eating.'
Score: 0.5 (captures correct predicted outcome but uses incorrect variable)

True: 'Users mentioning outdoor activities prefer healthy food.'
Generated: 'Users mentioning outdoor activities prefer healthy food.'
Score: 1.0 (perfect match)

Scoring scale:
- 1.0: Perfectly matches all variables and relationships
- 0.75: Captures primary relationship correctly but misses minor details
- 0.5: Partially correct (correct relationship or correct outcome, but missing important variables or conditions)
- 0.25: Minimal correct alignment (barely relevant but somewhat aligned in intent)
- 0.0: Incorrect, contradictory, or invalid/empty hypothesis

User Prompt

Evaluate how correctly the generated hypothesis captures the relationships described in the true hypothesis.

True Hypothesis: <hyp_true>
Generated Hypothesis: <hyp_gen>

Provide only the numerical score (0, 0.25, 0.5, 0.75, or 1.0).

---

Hypothesis Evaluation: Relationship Correctness

## B   Dataset Creation Details

### B.1   Paper Citation

**Prediction Target**   We consider the binary prediction target of highly cited papers. We group papers published in the same venue and year into a cohort. A paper $i$ published in year $t$ is considered highly cited ($C_{i,t,n,\alpha} = 1$) if, after $n$ years, its citation count is within the top $\alpha$ percent of its cohort, and $C_{i,t,n,\alpha} = 0$ if it is within the bottom $\alpha$ percent. By focusing on the most and least successful papers, we aim to maximize the potential signal in the differences between these two classes, making the distinguishing patterns more salient for abductive reasoning algorithms.

**Input Abstract**   There are many possible input features. We can divide them into (1) data: the paper itself, and (2) meta-data: descriptors of the paper such as publication venue, author affiliation, citation networks, etc. We choose to use only the paper text. Our aim is to generate hypothesis to understand how the content itself can influence impact. We only include the abstract instead of the full paper because (1) the

| Dataset | IND Split | OOD Split |
|---|---|---|
| Deception Detection | 800 truthful + 800 deceptive hotel reviews from original sources (Mechanical Turk + web) | 640 hotel reviews from four different cities and different web sources |
| AI-generated Content Detection | GPT-generated + human-written stories (for GPTGC); Llama-generated + human-written stories (for LlamaGC) | Cross-model: LlamaGC dataset for GPTGC OOD and vice versa |
| Persuasive Argument Prediction | Split by text corpus source | |
| Mental Stress Detection | Split by subreddit community | |
| News Headline Engagements | Headlines published before Dec 31, 2013 | Headlines published between Jan 1, 2014 and Nov 16, 2014 |
| Retweets | Tweets posted before Dec 23, 2011 | Tweets posted between Dec 23, 2011 and Oct 20, 2013 |
| Paper Citations | Papers from Health Affairs, Radiology, and NeurIPS published 2010–2016 | Papers from the same venues published 2012–2022 |

Table 9: IND and OOD splits for all real-world datasets in HypoBench. Each split is designed to test generalization across different domains, time periods, or data sources.

full paper has information beyond text, and would make it infeasible for LLMs and (2) current LLMs still have limitations on context length, making abstract more practical for various induction reasoning algorithms.

**OpenAlex API**   We build the citation dataset using the OpenAlex API (Priem et al., 2022). We noticed that the OpenAlex database is often unreliable, returning incomplete abstracts and short comentaries that are not full papers. We implemented mechanisms to automatically filter and clean data, which resulted in a total of 5324 data points.

**Journal Selection**   We select three journals according to the following principles:

- The journals should be among the most influential in their respective fields, as ranked by their impact factors from the Observatory of International Research (OOIR)[3]. The impact factor is chosen because it remains one of the most widely recognized indicators of journal influence. By targeting journals with high impact factors, we aim to minimize biases arising from suspicious citation practices, such as citation cartels or citation boosting services commonly found in lower-quality venues Ibrahim et al. (2024).

- The selected journals should represent three distinct academic fields, sufficiently distant from each other and with varying degree of distance. This selection strategy allows us to robustly assess the out-of-distribution (OOD) generalization capabilities of various methods.

- Each chosen journal must have a sufficient number of valid abstracts ($\geq 200$), as determined after filtering results from the OpenAlex API. The filtering process excludes non-article content such as commentaries, editorials, and incomplete abstracts, which are occasionally retrieved by the OpenAlex API.

**Difficulty Controls**   There are several parameters that can be used to control the difficulty of the dataset.

---

[3]https://ooir.org/index.php

| Journal | 2010–2016 | 2012–2022 |
|---|---|---|
| Health Affairs | 306 | 238 |
| Radiology | 490 | 328 |
| Conference on Neural Information Processing Systems | 386 | 538 |

Table 10: Number of data points by journal and time range.

- $n$ years after publication: The larger the $n$, longer the time span for the papers to accumulate citations, and thus would make it clearer which papers are truly high impact, increasing the signal to noise ratio. For results in Table 10, we used $n = 2$

- $\alpha$ percent of top and bottom papers to include: The smaller the $\alpha$, the greater the gap between the citation counts of the high and low impact papers, increasing the singal to noise ratio. For results in Table 10, we used $\alpha = 0.1$

## B.2 Marine Ecosystem

We choose one dataset from the marine biology domain in DiscoveryBench (Majumder et al., 2024) and convert it to our desired format. Most of the marine biology datasets are constructed based on the assumption that marine ecosystems have a complex combination of environmental factors that affect the target variable of interest, so we manually found one dataset with the groundtruth hypothesis "The average daily sunlight hours at a marine location increase as water clarity improves, particularly when there are fewer clouds." We set daily sunlight hours as our prediction target and keep all features about the marine ecosystem. As a result, there are a lot of noise variables (14 in total) not related to the groundtruth hypotheses. Since the daily sunlight hours is a floating point number and DiscoveryBench mostly consists of regression tasks, we use mean squared error (MSE) to measure the performance of different methods. Results on this dataset can be found in Table 11.

## B.3 Dataset Generation for Presidential Election and Personality Prediction

**Overview** To facilitate rigorous evaluation of machine learning models in a controlled experimental setting, we introduce a synthetic dataset generation pipeline that systematically constructs textual data with precisely defined labels. This dataset is designed to capture structured semantic variations through the integration of template-based text generation and feature-driven label assignment.

**Dataset Structure** The dataset consists of textual samples constructed by combining pre-defined templates with variable placeholders (blanks) and corresponding feature sets. These features introduce semantic differences, with their presence determining the final assigned labels. The overall process ensures a diverse and structured dataset while maintaining grammatical and semantic coherence. The key components of the dataset generation are:

- **Templates**: Serve as textual frameworks containing blanks to be filled with features.

- **Features**: Represent semantic variations that fill gaps with templates, influencing the final classification labels.

- **Labels**: Determined based on the occurrence of features and a randomly initialized multinomial logistic regression model.

**Text Generation Process** All textual components—including labels, templates, and features—are generated through large language models (LLMs). Labels are produced by prompting the LLM with user-defined task descriptions, ensuring they are semantically meaningful. Feature generation occurs in two stages: first, the LLM identifies the blank types required within a template based on task descriptions and labels; then, the LLM generates feature values for each blank type, ensuring their relevance to the assigned labels. Templates are then created based on task descriptions, labels, and feature types, with blanks designed to seamlessly integrate different feature variations while maintaining fluency.

As each feature type has multiple corresponding features, and each template contains multiple feature types as blanks, an exhaustive enumeration of all possible combinations is conducted. Each feature type is substituted with all possible feature values in its category, and each template undergoes substitution with all feature value combinations. This results in a comprehensive dataset capturing all possible feature interactions, ensuring diversity for model evaluation.

**Label Assignment Mechanism**  Labels are assigned through a multinomial logistic regression model with randomly initialized numeric weights. The input representation for the model is a boolean vector, where each dimension corresponds to a specific feature. A value of 1 indicates the presence of the feature in the text instance, while a value of 0 denotes its absence. This structured representation allows for systematic label assignment based on predefined logistic regression model. The multinomial logistic regression model is assigned with the randomly generated weight matrix with shape [num_classes, num_features]. The weight matrix is applied to input feature vectors to determine the impact of each textual feature on class assignment.

The label preference sorting operation is then applied independently for each feature across all classes in the weight matrix, yielding a ranked preference of classes based on their weight magnitudes. A positive weight increases the probability of assigning a feature-containing text to the corresponding class, while a negative weight decreases this probability. This process ensures that feature-class relationships are systematically captured and interpreted. The class-preference ranking obtained from the logistic regression weight matrix is translated into natural language explanations, providing an interpretable mapping between textual features and label assignments.

**Ground Truth Hypothesis Generation**  Ground truth hypotheses are generated based on the class preference ranking derived from logistic regression model weights. Specifically, the model weights per feature indicate the importance of each feature for class assignment. For each feature, we extract the maximum and minimum weights across classes to identify the most likely and least likely classes when a feature is present. These relationships are then converted into natural language hypotheses. For instance, a positive weight indicates that texts containing a particular feature are likely to be assigned to a given class, whereas negative weights suggest a reduced likelihood. By systematically translating these relationships into natural language statements, the dataset provides interpretable ground truth hypotheses clearly connecting textual features and their label assignments.

**Difficulty Settings**  To accommodate various levels of complexity, the dataset includes six predefined difficulty levels:

- **Level 0**: A single randomly selected feature determines the label; all texts containing this feature are assigned to one class, while others belong to a different class.

- **Level 1**: A single feature type influences classification, with all features of this type assigned nonzero logistic regression weights, while other feature types have no impact.

- **Level 2**: Three feature types contribute to label generation, while two additional feature types act as distractors with zero impact.

- **Level 3**: Introduces 10% label noise on top of Level 1.

- **Level 4**: Based on Level 2, with 25% of logistic regression weights randomly dropped.

- **Level 5**: Combines Level 2 conditions with 10% label noise and 25% logistic regression weight dropout.

Each difficulty level is available in two data presentation modes:

- **Regular / With Subtlety** Features are embedded into natural language templates, resulting in fluent and varied textual inputs. This setting simulates realistic linguistic variation and encourages models to generalize beyond surface patterns. It is well-suited for evaluating language understanding under more naturalistic conditions.

- **No Subtlety** Inputs consist of explicit enumerations of the features present in each instance, with no templated language. This setting eliminates linguistic variation, offering a controlled environment where the relationship between features and labels is made explicit. It supports fine-grained interpretability, simplifies error analysis, and isolates the impact of feature-based reasoning. Due to the lack of prompt-based augmentation, this variant is smaller in size but more deterministic in structure.

Combining the six predefined difficulty levels with the two presentation modes results in a total of 12 distinct datasets.

**Contrastive Difficulty Settings for Controlled Experiments** In addition to the predefined difficulty levels, we also introduce *contrastive difficulty settings* explicitly designed for controlled experimentation through systematic variation of three hyperparameters:

1. **Number of Features per Template**: Varies across the set $5, 10, 15, 20$, determining the complexity and semantic richness of the textual instances.

2. **Label Noise Ratio**: Defined as the proportion of randomly flipped labels, systematically adjusted through the values $0, 0.1, 0.2, 0.3$ to evaluate model robustness to labeling errors.

3. **Weight Dropout Probability**: Applied to randomly eliminate portions of logistic regression weights, varied over the range $0, 0.1, 0.2, 0.3$ to assess sensitivity to incomplete or noisy feature-class relationships.

By exhaustively combining these parameters, we construct a comprehensive grid search resulting in a total of 64 distinct dataset configurations. This structured approach enables fine-grained analysis of model performance under varying conditions of semantic complexity, labeling uncertainty, and structural ambiguity.

**Dataset Splitting, Formatting, and Conversion** The final dataset undergoes a standard train-validation-test split, where 70% of the data is allocated for training, 10% for validation, and 20% for testing. The dataset is then converted into **Hugging Face Dataset format**, ensuring compatibility with modern deep learning frameworks for streamlined experimentation.

### B.4 Dataset Generation for College Admission and Shoe Sales

For the college admission and shoe sales datasets, we use decision trees as the underlying model. We provide the dataset details below.

**College Admission.** For the college admission datasets, we include one base difficulty configuration and three different levels on four difficulty controls, including number of features, decision tree depth, noise level in outcome, and number of distractors. For base level, we only include one feature, tree depth one, no noise in outcome, and no distractors. We provide the detailed configurations for the other levels below:

- Number of features: 1, 5, 10, 15

- Tree depth: 1, 2, 3, 4

- Noise in outcome: 0%, 10%, 20%, 30%

- Number of distractors: 0, 3, 6, 10

The inputs to the college admission task will be a list of the candidate student's info, including the required features for each configuration. For example with the 5-feature configuration, we include the student's Math grade, English grade, number of publications, strong extracurricular activities, and recommendation letters. Tree depth, noise in outcome, and number of distractors will affect the underlying decision tree's decision

rules accordingly. Furthermore, for each college admission dataset, we construct a counterpart containing counterintuitive hypotheses, e.g., *"Students with an F grade in Math will be admitted."* These counterintuitive datasets enable additional evaluation of models and hypothesis generation methods in scenarios where prior knowledge is misleading or unhelpful.

**Shoe Sales.**   The shoe sales dataset has three variants. The first is generated by a one-level decision tree, and the other two are generated with two-level decision trees with different branching variables.

The input examples in all variants follow the format of "a [age] and [height] [gender] with [hat color] hat, [shirt color] shirt, and a [bag size] [bag color] bag." The goal is predict the color of shoes they will buy. Each feature is a categorical variable.

- Age (2): young / old

- Height (2): tall / short

- Gender (2): man / woman

- Hat color (6): red / orange / green / blue / black / white

- Shirt color (6): red / orange / green / blue / black / white

- Bag size (2): large / small

- Bag color (6): red / orange / green / blue / black / white

- Shoe color / label classes (6): red / orange / green / blue / black / white

We will publicly release all datasets in HYPOBENCH. More dataset details will be included in the official release.

## B.5   Synthetic Dataset Generation Examples

In this section we show some example dataset generation pipeline for the presidential election task. The personality prediction datasets are generated using the same framework, and we will release the detailed code in the official release of HYPOBENCH.

---

Given a tweet, determine the likely voting preference of the person for the 2024 U.S. presidential election. The classification should consider whether the individual is likely to vote for the Democratic candidate, the Republican candidate, a third-party candidate, or abstain from voting. The analysis should take into account explicit endorsements, political ideology, sentiment toward candidates and policies, use of partisan language, engagement with political topics, and references to past voting behavior. Additionally, indirect indicators such as reactions to major political events, stance on key social and economic issues, and alignment with party-affiliated hashtags or slogans should be factored into the prediction. The classification should aim to capture both strong political affiliations and nuanced, context-dependent voting tendencies.

---

Election Task Description

**Label Generation**   We applied the prompt *Synthetic Dataset Label Generation* from Appendix A.2, using the provided task description, and requested the LLM to generate three labels. The resulting labels generated by the LLM are:

- Likely Democratic Voter

- Likely Republican Voter

- Likely Third-party/Abstain Voter

**Feature Types Generation**  We employed the prompt *Synthetic Dataset Template Blank Type Generation* from Appendix A.2, leveraging the provided task description and the previously generated labels, requesting the LLM to generate four phrase types. The LLM generated the following phrase types:

- political_endorsement

- policy_stance

- partisan_language

- political_event_reaction

**Feature Generation**  Following label and feature-type generation, we prompted the LLM using the formatted *Synthetic Dataset Feature Generation* template from Appendix A.2. We provided the task description, generated labels, and feature types, and requested five phrases per feature type. The LLM-generated features for each type are:

- political_endorsement

    1. endorses the Democratic candidate
    2. champions conservative values
    3. advocates for third-party alternatives
    4. criticizes mainstream political parties
    5. promotes non-voting as a protest

- policy_stance

    1. advocates for universal healthcare
    2. opposes tax cuts for corporations
    3. supports immigration reform
    4. endorses climate change initiatives
    5. champions gun rights

- partisan_language

    1. promotes universal healthcare
    2. defends Second Amendment rights
    3. advocates for libertarian policies
    4. criticizes two-party system
    5. supports social justice initiatives

- political_event_reaction

    1. criticizes Supreme Court decision favoring conservatives
    2. praises Biden's climate change policy
    3. condemns government shutdown orchestrated by Republicans
    4. expresses frustration over lack of third-party debate presence
    5. celebrates passage of bipartisan infrastructure bill

**Templates Generation**  Using the LLM-generated feature types, we requested the LLM to produce textual templates with placeholders for feature insertion. Utilizing the *Synthetic Dataset Templates Generation* template from Appendix A.2, we asked for four templates aligned with the provided task description and labels. The resulting templates are:

- I'm planning to vote for [political_endorsement] because I strongly support their stance on [policy_stance]. This is especially important to me following [political_event_reaction], and I think it's critical that we all use our voices. I know some might disagree, but I can't stand the [partisan_language] being thrown around these days.

- After [political_event_reaction], I've been re-evaluating my stance on [policy_stance]. While I usually align with [political_endorsement], I find the [partisan_language] in current discourse off-putting. I'm not sure what my vote will be yet, but these issues are at the forefront of my mind.

- Despite the [partisan_language] I've seen, my vote is going to [political_endorsement] this election. Their position on [policy_stance] resonates with me, especially in light of [political_event_reaction]. It's crucial that we look beyond the rhetoric and focus on real issues.

- I was quite taken aback by [political_event_reaction], which led me to reconsider my position on [policy_stance]. The [partisan_language] makes it difficult to stay neutral, but I'm leaning towards [political_endorsement] as the election approaches.

**Grammar Enhancement for Features** Having generated all requisite textual components for the dataset, we requested the LLM to enhance grammatical coherence for each feature, ensuring seamless integration into the textual templates. We employed the *Synthetic Dataset Feature Grammar Enhancement* template from Appendix A.2 for this purpose.

**Ground Truth Hypothesis Generation** Ground truth hypotheses are generated systematically by leveraging the weights derived from the multinomial logistic regression model. Specifically, given a weight matrix of shape [num_classes, num_features], we analyze each feature independently across classes to determine its influence on label probabilities. For each textual feature, we compute the maximum and minimum weights across all classes, identifying the most and least likely class assignments respectively. Based on the polarity of these weights (positive, negative, or neutral), the hypotheses explicitly state the likelihood of texts containing specific features being assigned to certain classes. Each hypothesis follows the structured textual format:

```
If the "<feature_type>" of the given tweet is "<feature_value>", then it is <likelihood_1> to be classified as "<label_1>" and
<likelihood_2> to be classified as "<label_2>".
```

The categorization of likelihood terms follows these criteria:

- A positive weight implies a text with the corresponding feature is *likely* to belong to the associated class.

- A negative weight implies the feature-containing text is *unlikely* to belong to that class.

- A zero weight indicates *neutrality*, meaning the feature has no effect on class assignment probability.

- If both weights associated with a feature are negative or positive, the hypotheses differentiate levels of likelihood with terms such as *highly likely/unlikely* or *a bit likely/unlikely* based on the relative magnitude of the weights.

This structured process ensures interpretability and clarity in the feature-to-label mappings, aiding researchers in understanding and evaluating model performance.

## C   Additional Results

### C.1   Marine Ecosystem Results

Due to limited time and computational resources, we only have results for GPT on the marine ecosystem dataset (see Table 11). ITERATIVE REFINEMENT and HYPOGENIC achieves the lowest errors, indicating that updating and refining the hypotheses help predict the sunlight hours.

| Method | MSE | FDR | RC | HDR |
|---|---|---|---|---|
| Zero-shot inference | 1.790 | - | - | - |
| Few-shot inference | 0.528 | - | - | - |
| Zero-shot generation | 0.306 | 1.0 | 0.625 | 0.625 |
| IO PROMPTING | 0.448 | 1.0 | 0.625 | 0.625 |
| ITERATIVE REFINEMENT | 0.214 | 1.0 | 0.5 | 0.5 |
| HYPOGENIC | 0.275 | 1.0 | 0.5 | 0.5 |

Table 11: GPT results for different methods on the marine ecosystem dataset. MSE is measures the mean squared error after normalizing both predictions and labels using min-max scaling, where min and max are based on values of the target variable in the training data.

The hypothesis discovery rate, on the other hand, show different trends. All four hypothesis generation methods are able to find all the true features, as indicated by their FDR. So RC is the sole determiner of the HDR. After manually looking at the RC values for hypotheses, we find it hard to quantify for this task. So the HDR value is less accurate than MSE in reflecting the quality of generated hypotheses for the marine ecosystem dataset.

## C.2 IND Results on Real Datasets

| Method | GPT | | Qwen | | Llama | | DeepSeek | |
|---|---|---|---|---|---|---|---|---|
| | Accuracy | F1 | Accuracy | F1 | Accuracy | F1 | Accuracy | F1 |
| Zero-shot inference | 62.2 | 56.8 | 61.0 | 55.7 | 66.0 | 62.2 | 61.7 | 56.2 |
| Few-shot inference | 64.4 | 61.5 | 67.1 | 65.8 | 70.9 | 69.2 | 62.2 | 59.0 |
| Zero-shot generation | 61.4 | 56.6 | 61.8 | 56.1 | 62.4 | 56.6 | 61.6 | 56.1 |
| LITERATURE-ONLY | 60.9 | 56.3 | 60.1 | 53.7 | 60.9 | 53.6 | 58.4 | 51.6 |
| IO PROMPTING | 62.4 | 59.0 | 72.6 | 72.3 | 67.1 | 65.3 | 61.2 | 60.1 |
| ITERATIVE REFINEMENT | 63.7 | 61.2 | 71.3 | 70.6 | 70.1 | 69.7 | 59.8 | 59.7 |
| HYPOGENIC | 67.8 | 66.4 | 72.9 | 72.3 | 72.8 | 71.5 | 66.7 | 65.0 |
| LITERATURE + DATA | **71.9** | **71.3** | **76.0** | **75.7** | **75.2** | **74.3** | **73.2** | **72.5** |
| Finetuned Llama (Oracle) | IND Accuracy: 84.7 / F1: 84.7 | | | | | | | |

Table 12: IND Accuracy and F1 scores for different methods across models on real-world datasets

| Generation \ Inference | GPT | Qwen | Llama | DeepSeek |
|---|---|---|---|---|
| GPT | 71.93 | 65.62 | 62.43 | 63.57 |
| Qwen | 64.14 | 75.95 | 70.43 | 71.67 |
| Llama | 63.66 | 71.33 | 75.21 | 70.14 |
| DeepSeek | 64.14 | 71.52 | 69.95 | 73.24 |

Table 13: Cross-model hypothesis-based inference accuracy (%) for in-distribution (IND) data, using hypotheses generated from LITERATURE + DATA. Each row indicates the generation model and each column the inference model.

In Table 12, we include the full accuracy and F1 scores of all models and methods on the IND part of the real datasets. We observe similar trends as the results on the OOD part. Here, Qwen is still the best model, outperforming other models by 2.49% on average accuracy. We also see that LITERATURE + DATA is the best hypothesis generation method. On the other hand, finetuned Llama outperforms all other models and methods, outperforming Qwen with LITERATURE + DATA by 8.72%. This may suggest that there is still room for explaining more of the IND datasets.

We also report the cross model inference performance of the IND datasets in Table 13. The results reveal that Qwen, Llama, and DeepSeek are able to use the generated hypotheses from the other models in this

subgroup effectively. In contrast, GPT generated hypotheses are not as effective when given to the other models, and GPT is not able to effectively use the other models' generated hypotheses for inference.

## C.3 Additional Results on Synthetic Datasets

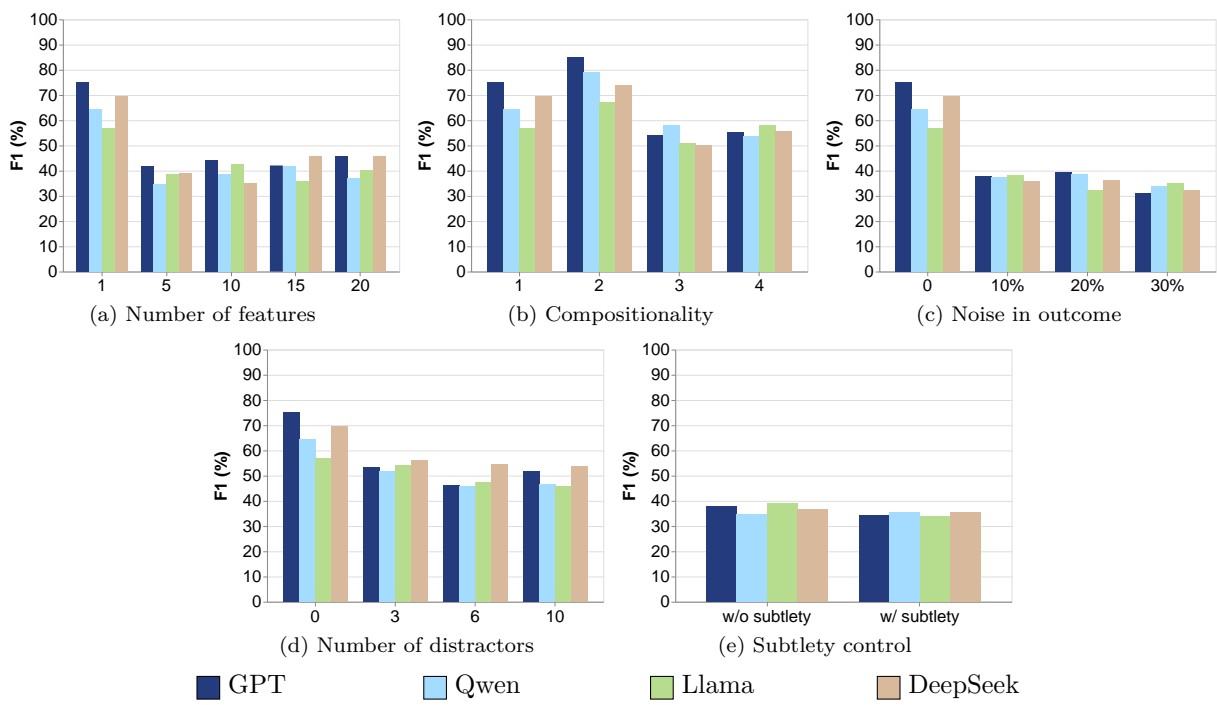

Figure 5: HYPOGENIC F1 scores on synthetic datasets with different task difficulty.

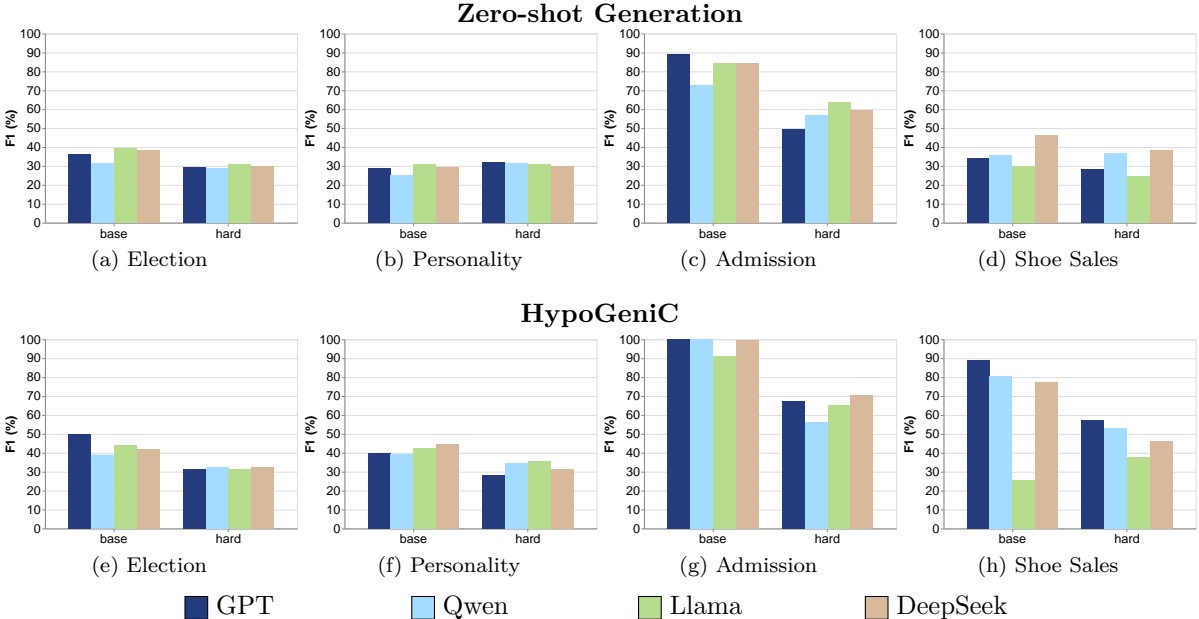

Figure 6: F1 scores of Zero-shot Generation and HYPOGENIC on four different synthetic datasets: Presidential Election, Personality Prediction, College Admission, and Shoe Sales. The results show that model priors can effect the quality of the generated hypotheses in different datasets.

| Method | GPT | | | Qwen | | | Llama | | | DeepSeek | | |
|---|---|---|---|---|---|---|---|---|---|---|---|---|
| | FDR | RC | HDR | FDR | RC | HDR | FDR | RC | HDR | FDR | RC | HDR |
| Zero-shot generation | 0.71 | 0.27 | 0.21 | 0.71 | 0.29 | 0.24 | 0.60 | 0.23 | 0.18 | 0.49 | 0.17 | 0.14 |
| IO PROMPTING | 0.81 | 0.36 | 0.30 | 0.79 | 0.37 | 0.31 | 0.82 | 0.35 | 0.30 | 0.81 | 0.35 | 0.29 |
| ITERATIVE REFINEMENT | 0.70 | 0.25 | 0.18 | 0.82 | 0.22 | 0.19 | 0.82 | 0.24 | 0.21 | 0.68 | 0.24 | 0.19 |
| HYPOGENIC | 0.95 | 0.48 | 0.46 | 0.93 | 0.43 | 0.41 | 0.98 | 0.45 | 0.44 | 0.96 | 0.45 | 0.44 |

Table 14: Hypothesis discovery rates for all model and methods. We report feature discovery rate (FDR), relationship correctness (RC), and the final hypothesis discovery rates (HDR). The results are averaged across all difficulty configurations.

| Method | GPT | | Qwen | | Llama | | DeepSeek | |
|---|---|---|---|---|---|---|---|---|
| | Accuracy | F1 | Accuracy | F1 | Accuracy | F1 | Accuracy | F1 |
| Zero-shot inference | 36.2 | 32.8 | 35.2 | 32.5 | 36.4 | 33.3 | 34.9 | 32.9 |
| Few-shot inference | 34.2 | 32.2 | 36.4 | 34.3 | 36.7 | 34.7 | 39.5 | 37.6 |
| Zero-shot generation | 37.4 | 33.9 | 36.8 | 32.5 | 37.7 | 33.9 | 36.4 | 33.6 |
| IO PROMPTING | 42.7 | 36.5 | 43.8 | 36.9 | 44.1 | 37.6 | 39.6 | 35.0 |
| ITERATIVE REFINEMENT | 41.5 | 36.1 | 42.5 | 36.8 | 45.6 | 38.7 | 34.7 | 32.5 |
| HYPOGENIC | 50.0 | 42.8 | 48.9 | 41.4 | 47.3 | 40.0 | 49.2 | 41.8 |

Table 15: Accuracy and F1 scores for different methods across models on synthetic datasets. We report the average accuracy and F1 scores across all difficulty configurations.

In Table 15 and Table 14, we report the aggregated performance of all models and methods on the synthetic datasets across all configurations. The results reveal that GPT achieves the best performances in terms of both average HDR score and accuracy. Additionally, we see that HYPOGENIC outperforms all other hypothesis generation methods with a large margin. However, this aggregated results show that recovering the ground-truth hypotheses and fully explaining the synthetic datasets in HYPOBENCH remain challenging for all models, as the best model only achieves 50.02% on average accuracy and 46% on average HDR score. This result further highlight the value of HYPOBENCH as a resource to advance models and hypothesis generation problems.

### C.4   Evaluation with O3 Reasoning Model

A natural question is whether more powerful reasoning models can solve HYPOBENCH through superior reasoning capabilities. We evaluated OpenAI's O3 model, a state-of-the-art reasoning model with extended chain-of-thought capabilities, on a subset of 11 synthetic datasets spanning 4 task domains.

For each dataset, we provide O3 with the complete training data and instructed to generate exactly 20 hypotheses as a JSON list. This gives O3 a potential advantage over iterative methods like HYPOGENIC, as it sees the entire dataset at once.

We show the results in Table 16. O3 achieved an average HDR of 0.52, with near-perfect feature discovery (FDR=0.99) but moderate relationship correctness (RC=0.52). Performance varied significantly: O3 achieved perfect HDR on simpler tasks (admission/level_2, election/level0) but only 0.25 on complex tasks (admission/level_4, preference/level5). Compared to HYPOGENIC with GPT-4O-MINI, O3 outperformed on some tasks (admission/level_3: 0.45 vs 0.30) but underperformed on others (shoe_simple: 0.75 vs 0.88).

These results demonstrate that HYPOBENCH remains challenging even for state-of-the-art reasoning models, which highlights the value of HYPOBENCH as a resource for improving models' hypothesis generation capabilities.

| Dataset | FDR | RC | HDR |
|---|---|---|---|
| admission/level_2/depth_2 | 1.00 | 1.00 | 1.00 |
| admission/level_3/depth_3 | 1.00 | 0.45 | 0.45 |
| admission/level_4/depth_4 | 1.00 | 0.25 | 0.25 |
| election/level0_nosubtlety | 1.00 | 1.00 | 1.00 |
| election/level5_nosubtlety | 1.00 | 0.40 | 0.40 |
| preference/level0 | 1.00 | 0.50 | 0.50 |
| preference/level5 | 0.93 | 0.27 | 0.25 |
| shoe_two_level/simple | 1.00 | 0.75 | 0.75 |
| shoe_two_level/hard | 1.00 | 0.38 | 0.38 |
| **Average** | **0.99** | **0.52** | **0.52** |

Table 16: O3 performance on synthetic datasets. Despite near-perfect feature discovery (FDR), relationship correctness (RC) remains moderate, yielding 0.52 average HDR.

## D  Experiment Details

### D.1  Implementation Details

In this section, we report the implementation details of the selected hypothesis generation methods in HYPOBENCH.

**Zero-shot generation.**  As a baseline method for hypothesis generation, we prompt the LLMs directly with the task descriptions and instructions to generate relevant hypotheses, without providing any additional information. This method represents the models' existing knowledge and ability to extract useful hypotheses from it.

**Literature-based generation.**  Another baseline method we consider is to use existing literature for hypothesis generation. We adopt the method design from Liu et al. (2025) and curated the necessary data for running this method. We use their default hyperparameters and generation prompts for all the experiments in HYPOBENCH.

**IO Prompting and Iterative Refinement.**  Following Qiu et al. (2024), we re-implement the IO PROMPTING and ITERATIVE REFINEMENT method using the exact prompts and the reported best hyperparameters in the original paper. Specifically, we train for 3 epochs with 10 examples, generate 5 hypotheses in each iteration, and update the previous hypotheses with feedbacks.

**HypoGeniC and Literature + Data.**  We adopt the exact implementations for HYPOGENIC and LITERATURE + DATA in their official code release (Zhou et al., 2024; Liu et al., 2025). We also use the provided prompts and the reported best hyperparameters for all experiments. For both methods, we keep a hypothesis bank size of 20, initialize with 10 examples, and train with one epoch of 200 examples. With LITERATURE + DATA, we use 6 rounds for the hypothesis refinement process.

### D.2  Costs

**Costs for running real datasets.**  For running the complete pipeline of HYPOBENCH on one real dataset, which includes running all hypothesis generation methods, costs approximately $5.5 in total, and 4 hours using 4 NVIDIA A100s for each of Qwen-2.5-72B-Instruct, Llama-3.1-70B-Instruct, and DeepSeek-R1-Distilled-Llama-70B. The cost breaks down to approximately $1.5 for running all the hypothesis generation methods with GPT-4o-mini, and $1 each for running the qualitative ratings using GPT-4o for all four models.

**Costs for running synthetic datasets.**  For each of the synthetic datasets, the complete pipeline of HYPOBENCH costs $2 in total, plus 4 hours using 4 NVIDIA A100s for all four selected models. The cost further breaks down to approximately $1.2 for running the generation methods with GPT-4o-mini, and $0.2 for running the HDR evaluations using GPT-4o for all four models.

