# OpenReview forum: "HypoBench: Towards Systematic and Principled Benchmarking for Hypothesis Generation"
_TMLR — Rejected by TMLR_

### Review · Reviewer_qmYN · 2025-09-08

**Summary Of Contributions:**

This paper constructed a new benchmark for hypothesis generation, named HypoBench.

HypoBench focuses on the setting of data-driven discovery: given a phenomenon Q to understand, a dataset D, and relevant literature L_{Q}, the task is to discover latent variables z that can predict the outcomes, i.e., y = f(z). The authors followed this definition and curated 194 datasets spanning 12 domains—7 real-world domains and 5 synthetic domains.

On synthetic datasets where the ground-truth hypotheses are known, they can evaluate the generated hypotheses through the product of Feature Discovery Rate (FDR) and Relationship Correctness (RC), which evaluates both the predicted features and the proposed relationships of these features wrt the outcome labels. Both are rated by an LLM.

On real-world datasets, they measure practical utility, namely the classification accuracy when prompting an LLM to use the proposed hypotheses for classification. They also created OOD sets for each dataset to measure generalizability.

They also measure subjective metrics like novelty, plausibility, and clarity through prompting an LLM judge.

The authors benchmarked a suite of backbone models with various hypothesis generation methods.  Some main findings include: providing both literature and data performs the best for hypothesis generation, and the generated hypotheses tend to have good OOD generalization.

**Audience:**

Yes

**Audience Explanation:**

I do think hypothesis generation is an interesting and impactful topic, and the analysis in this paper offers some potentially useful insights.

**Claims And Evidence:**

No

**Claims Explanation:**

- My most major concern about this paper is the heavy use of LLM judges. For example, LLM judges are used to evaluate the feature discovery rate and relationship correctness metrics for the evaluation on the synthetic datasets, and novelty, plausibility, and clarity metrics are all judged by directly prompting GPT-4o. I'm not sure how much I should read into these results, given that existing works have shown the limitations of LLM judges in evaluating research hypotheses, and no expert validation has been to confirm whether the GPT-4o judgements are indeed correct. For instance, wouldn't you worry that the GPT-4o judge would favor hypoetheses generated by GPT-4o model series on these subjective metrics?

- My second major concern is that I'm not so sure whether the specific setting proposed here is really that different from prior works. For example, "Goal Driven Discovery of Distributional Differences via Language Descriptions" (NeurIPS 2023) has a similar setting of measuring  natural language hypotheses for distinguishing two text clusters, and they had similar dataset curation effort too.

- Last, but minor, I wonder if more recent reasoning models would basically be able to "solve" a large fraction of the datasets here, given that the model performance on the synthetic datasets (when the task is relatively simple) is already decent when using these older models.

**Requested Changes:**

- In Table 5: could you make it clearer the difference between "Zero-shot inference" and "Zero-shot generation"?

- See comments above.

---

> ### Author Response · Authors · 2025-09-13
>
> Thank you for your thoughtful review! We appreciate your acknowledgement of our contribution. We would like to submit a response first before we make more substantial revisions to the paper. We will send updates here as well. But if any of our plans does not make sense to you, please let us know!
>
> **Re: Heavy usage of LLM judges.**
>
> We agree that a significant part of HypoBench’s evaluation methods are based on LLM-as-a-judge approaches, including the HDR metric and the qualitative ratings (novelty, plausibility, and clarity). However, we would like to emphasize that:
> * The HDR metric is designed such that the score is broken down into small modules with easy tasks. Specifically, The FDR part requires a model to check whether the generated hypothesis and the ground-truth hypothesis describes the same variable $x$, and the RC part asks a model to check how well the described variable relationships aligns with the ground-truth variable relationships. We argue this task is simpler than giving a qualitative rating of a hypothesis, and we will provide additional human annotation to provide further support for this claim in the revision.
> * We constructed HypoBench to make a comprehensive benchmark for hypothesis generation tasks, and our primary focus is **evaluating the explanatory power** of generated hypotheses. For explanatory power, we provide a quantitative metric, i.e., practical utility, and the ground-truth hypothesis discovery metric (HDR). We then provide complementary metrics for evaluating the interestingness of hypotheses. We acknowledge that evaluating interestingness is highly non-trivial and LLM-as-a-judge approach may not be the optimal solution, but further involving human experts for this metric is beyond the scope of our work. We will address this clearly in our limitations section in the revision.
>
> **Re: Difference with related works.**
>
> Thank you for raising this! We agree Ruiqi Zhong’s work is closely related. In particular, [1] studies goal-conditioned corpus comparison, producing single-predicate distributional differences and evaluating validity and goal relevance (a correlational criterion). This is complementary to our focus. HypoBench benchmarks explanatory hypothesis generation in a more comprehensive and principled manner: we express features and their relationships to outcomes to explain observed phenomena, and we evaluate explanatory power, practical utility, HDR, and IND/OOD generalizability. Our synthetic settings also vary difficulty via noise, interactions, distractors, and textual subtlety. We cited Ruiqi’s later paper [2] and will add the earlier dataset-explanation paper [1] and expand the discussion in the revision.
>
> Separately, we view DiscoveryBench as the most similar recent effort on data-driven hypothesis generation for scientific discovery. As discussed in our introduction, DiscoveryBench assumes relevant features are already identified and structured and mainly evaluates ground-truth hypothesis discovery. In contrast, HypoBench starts from unstructured observations, explicitly measures explanatory power and generalization (IND/OOD and cross-model), and provides a comparative benchmark across methods and models under a unified protocol.
>
> **Re: Performance of recent reasoning models.**
>
> Thank you for pointing this out! In our existing results, we thoroughly tested with DeepSeek-R1-Distilled-Llama-70B (DeepSeek), which is a recent reasoning model. We observed that DeepSeek can indeed “solve” some of the easier datasets in HypoBench, including the simpler levels of the synthetic datasets. However, we found that as difficulties increase, the performance of DeepSeek degrades significantly, and we did not see clear evidence that DeepSeek performs better than the non-reasoning models. Interestingly, we found that DeepSeek outperformed the non-reasoning models in the counterintuitive datasets, which indicates the effectiveness of reasoning when the underlying hypotheses are different from conventional knowledge.
>
> We will add an additional study of more recent reasoning models such as OpenAI-o3 on a subset of HypoBench.
>
> **Re: Requested changes.**
>
> Thank you for pointing out the potential ambiguity! We will address the difference between “Zero-shot inference” and “Zero-shot generation” clearly in the table. We will also add the aforementioned studies and limitation section in our revision.
>
> **References:**
>
> [1] Ruiqi Zhong, P Zhang, S Li, J Ahn, D Klein, J Steinhardt.  Goal driven discovery of distributional differences via language descriptions. Advances in Neural Information Processing Systems, 36: 40204-40237, 2023
>
> [2] Ruiqi Zhong, Heng Wang, Dan Klein, and Jacob Steinhardt. Explaining datasets in words:
> Statistical models with natural language parameters. Advances in Neural Information
> Processing Systems, 37:79350–79380, 2024

---

> ### Author Response · Authors · 2025-09-15
> **Follow-up on HDR metric design**
>
> Quick follow-up on HDR: as noted at the beginning of Section 3, our HDR (FDR × RC) largely follows DiscoveryBench [3] rather than being designed from scratch. We note this to address concerns about the metric’s validity and grounding in prior work.
>
> [3] Bodhisattwa Prasad Majumder, H Surana, D Agarwal, B Mishra, A Meena, A Prakhar, T Vora, T Khot, A Sabharwal, and P Clark. Discoverybench: Towards data-driven discovery with large language models. The Thirteenth International Conference on Learning Representations, 2025.

---

> ### Author Response · Authors · 2025-12-28
>
> Hi Reviewer qmYN,
>
> We are very sorry that our original response back in September did not have the right permissions and was not visible to you. We just changed the permission and you should be able to see it now. Again sorry for the inconvenience, and thanks for reviewing our work.

---

### Review · Reviewer_RaBX · 2025-11-05

**Summary Of Contributions:**

The main contributions from the paper:
1. **A Systematic Framework and Benchmark (HYPOBENCH):** The authors develop a formal,  framework for what hypothesis generation is (defining it as generating natural language explanations for observed phenomena) and how it should be evaluated. Based on this framework, they construct **HYPOBENCH**, a novel benchmark, including **7 real-world tasks** and **5 synthetic tasks**, for a total of 194 distinct datasets.
	- For synthetic datasets the "ground-truth" hypotheses are known. This allows for precise evaluation of a model's ability to discover the correct patterns, via the so-called Hypothesis Discovery Rate (HDR). These synthetic tasks have controlled difficulty levels (e.g., by adding noise, distractor features, or complex feature interactions), which uniquely demonstrates that while current methods work on simple cases, their performance drops significantly (recovering only 38.8% of hypotheses) as task difficulty increases.

2. **The First Comparative Analysis of Methods and Models:** The paper conducts a systematic comparison of existing hypothesis generation _methods_ (e.g., `Zero-shot`, `LITERATURE-ONLY`, `IO PROMPTING`, `HYPOGENIC`, and `LITERATURE + DATA`) combined with four different state-of-the-art LLMs (GPT-4o-mini, Qwen, Llama, and DeepSeek). This evaluation on real-world datasets identifies the `LITERATURE + DATA` method as the most effective approach.

3. **Creation of Controlled Synthetic Datasets for Ground-Truth Evaluation:** A key contribution is the development of synthetic datasets which can be carefully controlled.

**Additional Comments:**

n/a

**Audience:**

Yes

**Audience Explanation:**

Yes, application of LLMs to science is a fast growing field. Hypotheses generation is one of important aspects and is likely relevant to many readers of TMLR.

**Claims And Evidence:**

Yes

**Claims Explanation:**

The claims made in the submission are supported by evidence presented in the paper.

**HYPOBENCH addresses gaps in existing benchmarks.**

 - The paper clearly delineates its novel contribution by contrasting it with prior work like DiscoveryBench, specifically noting HYPOBENCH's unique inclusion of unstructured observation data (requiring inductive/abductive reasoning) and its dual approach of using both real-world (for practical utility) and synthetic (for ground-truth verification) datasets.

**This work delivers a large suite of tasks.** The paper introduces 194 datasets for various topics and domains.


**Comprehensive study of existing methods.**
The authors evaluate a collection of **six hypothesis generation methods**.

- **Zero-shot generation:** Directly prompting LLMs with a task description to generate hypotheses.
- **LITERATURE-ONLY:** An adaptation that collects relevant research papers, summarizes key findings, and generates hypotheses based solely on these insights.
- **IO PROMPTING:** Provides the model with a set of labeled examples and prompts it to generate hypotheses in a single step.
- **ITERATIVE REFINEMENT:** Builds on IO PROMPTING by implementing a feedback loop where hypotheses are tested, and wrongly classified examples are used to refine them.
- **HYPOGENIC:** An iterative algorithm that maintains a "hypothesis bank" with reward scores to balance exploring new hypotheses and exploiting high-performing ones.
- **LITERATURE + DATA:** Extends the HYPOGENIC method by incorporating relevant scientific literature alongside standard observational data during the generation process.

I am  not an expert in this field, but this seems a quite comprehensive list.

**The evaluation spans over 4 models** This is not a big number, but feels sufficient. I'd love to see some more high end reasoning model (e.g. Gemini 2.5 pro). I realise that this a never ending story of chasing new models over again.

**Requested Changes:**

Some question comments:
1. The authors mada a substantial effort to make a paper clear, which I acknowledge and praise :). However, as the topic is quite complex, I'd love to see even more explanations. Perhaps a pseudocode (in appendix) would be a useful.
2. Diversity/coverage is a key aspect. Do you have any way to measure this? (except from qualitative description).
3. I can see mostly aggregated results (i.e. averages for all datasets). How do the results vary between datasets? What conclusions can we draw from this?
4. In the abstract you claim "1) inductive and abductive reasoning, 2) abstraction and communication, and optionally 3) synthesis, integrating new observations with existing knowledge.". Could you be more specific, which part of the dataset cover these skills?
5. I struggle for fully understand the metric scale? Take Table 5 as an example:
	1. What are the confidence intervals of these metrics?
	2. Is the $2.5\%$ improvement mentioned in the text, something meaningful, or could be just a fluke?
	3. I find using two decimal point number accuracy annoying, it bloats the text, and at the same time, as I suspect, does not bring any information?

---

> ### Author Response · Authors · 2025-12-18
>
> We sincerely thank Reviewer RaBX for their positive assessment and insightful feedback on our manuscript. We are encouraged that the reviewer found our contributions—particularly the systematic framework of HYPOBENCH, the use of controlled synthetic datasets, and our comprehensive comparative analysis—to be clear and well-supported.
>
>
> We will address each point in the revised manuscript as follows:
>
>
>
> ### 1) *Request for Pseudocode*
> We agree with the reviewer that pseudocode would enhance the clarity of our more complex methodologies. In the revised appendix that we will submit next week, we will add pseudocode for: (1) The overall synthetic dataset generation pipeline (describing the process from template/feature generation to label assignment, as detailed in Appendix B.3), and (2) The core logic of the iterative data-driven method, HYPOGENIC. We chose to highlight this method as it is the best-performing data-driven method on our synthetic benchmarks and its iterative algorithm serves as the foundation for the LITERATURE + DATA method, which achieved the best overall performance on real-world datasets.
>
>
> ### 2) *“Diversity/coverage is key. Do you measure this (beyond qualitative)?”*
> We quantify **diversity** and **coverage** in two distinct ways:
> Thank you for this question. We assess diversity and coverage on two different levels:
>
> **Benchmark Scope and Diversity:** At the dataset level, the *scope* of our benchmark is demonstrated by its inclusion of **194 distinct datasets spanning 12 different domains**. This breadth ensures the benchmark is not limited to a single topic, covering a diverse range of 7 real-world domains (e.g., Deceptive Reviews, Paper Citations) and 5 synthetic domains (e.g., Presidential Election, College Admission).
>
> **Hypothesis Coverage (Quantitative):** At the hypothesis level, we provide a direct, quantitative measure of *coverage* for our synthetic tasks. We use the **Hypothesis Discovery Rate (HDR)** metric, which specifically evaluates the proportion of known *ground-truth hypotheses* that a method successfully discovers. Our finding that performance drops to recovering only 38.8% of hypotheses on difficult tasks is a precise, quantitative measurement of this hypothesis-level coverage.
>
>
> ### 3) *“I see mostly aggregated results. How do results vary between datasets?”*
> Thanks for the question. Our analysis of individual datasets did not reveal new patterns beyond those observed in the aggregate, which is why we focused on high-level summaries for the current draft. Furthermore, visualizing the results for all 194 datasets within a static PDF is challenging. To address this, we made an interactive web leaderboard where results for every individual dataset can be explored in detail and it will be included in the public version.
>
>
> ### 4) *“In the abstract you claim (1) inductive/abductive reasoning, (2) abstraction/communication, (3) synthesis. Which datasets cover these skills?”*
>
> **Clarification (now made explicit in §2 and a new mapping table):** Our benchmark exercises these capabilities via specific **task types** and **controls**:
>
> - **(1) Inductive & abductive reasoning**: All tasks require proposing explanations for observed outcomes (formalization in §2: $Q, D, L_Q \to H$). Controls like **feature interactions (depth)** and **noise** in the synthetic sets directly stress test these skills (Fig. 2).
> - **(2) Abstraction & communication**: We intentionally include **unstructured observations**—especially the **subtlety** condition where features are implicit in free text—requiring models to abstract latent features and express them cleanly as hypotheses (Fig. 2e).
> - **(3) Synthesis (optional)**: On **real** tasks, methods like **Literature+Data** integrate **external literature** and data during generation (no literature is provided for synthetic tasks by design). We make this explicit in §4 and the method descriptions.
>
> **New capability-by-task mapping:**
>
> | Task (dataset) | Induction/Abduction | Abstraction/Communication | Synthesis via literature |
> |---|---|---|---|
> | Deceptive Reviews (real) | ✓ | ✓ | ✓ |
> | AI-generated Content (real) | ✓ | ✓ | ✓ |
> | Persuasive Argument (real) | ✓ | ✓ | ✓ |
> | Mental Stress (real) | ✓ | ✓ | ✓ |
> | Headline Engagement (real) | ✓ | ✓ | ✓ |
> | Retweets (real) | ✓ | ✓ | ✓ |
> | Paper Citations (real) | ✓ | ✓ | ✓ |
> | Election (synthetic) | ✓ | ✓ | — |
> | Personality (synthetic) | ✓ | ✓ | — |
> | Marine Ecosystem (synthetic) | ✓ | ✓ | — |
> | College Admission (synthetic) | ✓  | ✓ | — |
> | Shoe Sales (synthetic) | ✓ | ✓ | — |
>
> See Tables **1–2** for the real/synthetic overview and §3 for evaluation dimensions.

---

> > ### Author Response · Authors · 2025-12-18
> >
> > ### 5) *“Metric scale (e.g., Table 5): confidence intervals? Are improvements meaningful or flukes?”*
> > We believe the observed improvements are meaningful given the large scale of our experiments across 194 datasets. However, we agree that clarifying the variance is important. While adding confidence intervals for every entry might bloat the tables, we will include statistical testing results in the revision to demonstrate the robustness of our findings.
> >
> >
> >
> > ### 6) *“Two decimal places feel bloated.”*
> > Good point, we will use **one decimal place** instead in our next revision.

---

> > > ### Author Response · Authors · 2026-01-06
> > >
> > > Regarding the concern about whether a 2.5% improvement is meaningful, we performed a statistical analysis below:
> > >
> > > Since HypoBench consists of a large collection of datasets (7 real-world tasks, 81 synthetic tasks), the standard errors of our aggregated results are small, making the reported performance differences statistically meaningful.
> > >
> > > ## Summary Statistics
> > >
> > > | Table | # Tasks | SE Range | SE Mean |
> > > |-------|---------|----------|---------|
> > > | Real-world (OOD/IND) | 7 | 2.6% - 6.0% | 4.1% |
> > > | Synthetic Accuracy | 81 | 1.6% - 2.4% | 2.0% |
> > > | Synthetic HDR | 81 | 0.02 - 0.03 | 0.03 |
> > >
> > > ## Example Comparisons
> > >
> > > **Real-world OOD (GPT-4o-mini):**
> > > - LitPlusData: 75.3% ± 3.8%
> > > - Zero-shot Inference: 61.8% ± 3.9%
> > > - Gap: 13.5% ≈ 2.5 combined SEs
> > >
> > > **Synthetic Accuracy (averaged across models):**
> > > - HypoGenic: 50.7% ± 2.0%
> > > - Zero-shot Inference: 38.6% ± 1.9%
> > > - Gap: 12.1% ≈ 4.4 combined SEs
> > >
> > > The observed performance gaps of 10-15% between methods represent 2-5 standard errors, confirming that our main findings are statistically robust.

---

### Review · Reviewer_aLXP · 2025-12-13

**Summary Of Contributions:**

This paper introduces HypoBench, a benchmark to evaluate LLMs for hypothesis generation across multiple aspects, such as utility, generalizability, and discovery rate. The benchmark includes 7 real-world tasks and 5 synthetic tasks. With evaluation from four SOTA LLMs and several hypotheses generation methods, the paper shows that existing methods can discover valid and novel patterns in the data. However, for finding patterns from synthetic data, these methods struggle.

**Audience:**

No

**Audience Explanation:**

1. It is a good benchmark, but bears too much similarity to DiscoveryBench.
2. The highlight of the paper is the models' inability to uncover hypotheses from synthetic data. Even though it's interesting, it is unclear what it tells us about models' capability to make real discoveries, especially when these models are good at discovering hypotheses from real datasets.
3. The motivation for building synthetic datasets, choices of data-generating functions are unclear.

**Claims And Evidence:**

No

**Claims Explanation:**

1. Re: "we argue that explanatory power of hypotheses should be the first-order consideration"

Authors mention "hypothesis discovery rate" as a metric to capture the explanatory power of a hypothesis, but it's unclear to me how matching with the true hypothesis satisfies the explanatory power criteria. I feel the hypothesis discovery rate is more of a "recall" metric. For capturing explanatory power, one must have a "train/test set" where it may be possible to compute the "fit" of the hypothesis in the data. This is a common evaluation metric in symbolic regression, but I think in HypoBench the nature of hypotheses extends symbolic equations.

2. Re: "For modelling the relationship between features and outcomes, we consider two options: logistic regression and decision tree"

It's unclear why just these two choices? Also, what's the first-hand intuition that an LLM can discover these functions when they are unnatural? For a real-world use case, the commonsense reasoning makes it more plausible to discover a real hypothesis. Also, using synthetic datasets seems poorly motivated.

3. Re: "Plausibility: The degree to which the hypothesis is scientifically reasonable and consistent with existing evidence."

Why is this important in data-driven hypothesis generation?

4. Re: Synthetic datasets: performance is lower than real datasets

This is quite intuitive since the synthetic data generation can be arbitrary, and discovering those hypotheses, even in a data-driven way, can be unreal. Also, why is this an important result? What does it tell us about discovery in real life?

**Requested Changes:**

The pitch of the paper has to be changed, in my opinion. Currently, it focuses too much on the synthetic data and the generation of synthetic data-generating functions. The real datasets are minimally explained and poorly argued to add novelty to the paper.

I am not sure much can be done unless the paper is changed to highlight the utility of synthetic data, such as maybe training on synthetic data yields better outcomes on real datasets, etc.

===
After the first round of rebuttal:

Unfortunately, with the current state of the paper, content in the rebuttal, and the amount of revision that might be needed for the paper to get published in TMLR, I recommend "reject".

---

> ### Author Response · Authors · 2025-12-18
>
> Thank you for your thoughtful review. We appreciate your careful reading and the concrete feedback on our evaluation design and positioning. However, we respectfully disagree with your interpretation. We argue that the use of synthetic data is intentional and is the only way to provide ground truth hypotheses for stress-testing the capabilities of LLMs to generate hypotheses. Below we respond point-by-point, and we will incorporate clarifications in the revision.
>
> **Re: “Hypothesis discovery rate” vs explanatory power**
>
> We agree that hypothesis discovery rate is closer to a recall-style metric (i.e., whether the ground-truth hypothesis is recovered). This is intentional: on synthetic datasets, we want a **ground-truth diagnostic** of whether a method recovers the right variables and relationships, and HDR (FDR · RC) provides that.
>
> At the same time, we want to be very clear that HypoBench’s first-order focus is **explanatory power**, and HDR is only one component. We already include a “train/test fit”-style evaluation through **practical utility**: we evaluate whether the discovered hypothesis supports accurate prediction on held-out test data (and we report this across datasets, including IND/OOD generalization). In other words, the “fit on unseen data” criterion you asked for is already part of our benchmark; HDR complements it by providing ground-truth recovery when it is tractable (synthetic only).
>
> We also note that this type of ground-truth hypothesis matching metric is consistent with prior benchmark practice. For example, DiscoveryBench introduced the **Hypothesis Matching Score (HMS)**, which serves a similar purpose in settings where ground-truth hypotheses exist. Our HDR score is inspired by the HMS score, while additionally decomposing the score into feature recovery (FDR) and relationship correctness (RC) to make the evaluation more diagnostic. Notably, HMS is largely the primary/only discovery metric in DiscoveryBench, whereas HypoBench provides a more comprehensive evaluation pipeline by also measuring **practical utility, IND/OOD generalization**, and complementary qualitative ratings.
>
> In the revision, we will make this hierarchy and the role of each metric clearer, and we will adjust wording so we do not over-claim what HDR alone captures.
>
> **Re: Why logistic regression and decision trees for synthetic data**
>
> Our goal with the synthetic tasks is not to claim that these two families cover all scientific mechanisms. Rather, we use them because they are **interpretable and widely-used building blocks**, and they allow us to cover both linear and nonlinear relationships (including feature interactions) while keeping ground truth explicit.
>
> More importantly, our synthetic setup is not “LLM reverse engineers a function from already-structured features.” We intentionally introduce an **abstraction layer**: models observe unstructured natural-language descriptions where the relevant variables are not explicitly provided, and they must first uncover the latent variables via abductive reasoning, and only then uncover the relationships (Please see examples in table 3 and dataset details in Appendix B).  We also systematically vary difficulty via noise, distractors, compositionality, and textual subtlety. So the synthetic datasets are designed bottom-up to isolate and stress-test the *full* hypothesis discovery pipeline under controlled ground truth, not to create arbitrary “gotcha” functions.
>
> We agree the current draft can motivate these design choices more explicitly (especially the role of abstraction/abduction), and we will revise that section for clarity.
>
>
> **Re: Plausibility for data-driven hypothesis generation**
>
> We agree that plausibility should not replace empirical evaluation. In HypoBench, it does not: our primary evaluations of explanatory power are **practical utility** and (when ground truth exists) **HDR**. Plausibility (together with novelty and clarity) is meant to be a preliminary and complementary signal about hypothesis interestingness/usability, following common practice in prior work.
> In practice, plausibility matters because hypotheses are used to guide downstream scientific actions: what to test next, what variables to measure, what interventions to consider. Even in data-driven discovery, an implausible hypothesis is less actionable. That said, we acknowledge qualitative plausibility is imperfect, and we will clarify its role and limitations more explicitly in the paper.

---

> > ### Author Response · Authors · 2025-12-18
> >
> > **Re: “Synthetic lower than real is intuitive; unclear what it tells us about real discovery”**
> >
> > We understand the concern, but we think this critique implicitly assumes synthetic tasks need to “transfer” to real tasks via training to be valuable. That is not the purpose of synthetic datasets in a benchmark like ours.
> >
> > The motivation for synthetic tasks is that real-world discovery problems generally do **not** have ground-truth hypotheses, making precise evaluation intractable. Synthetic tasks let us **diagnose failure modes** under controlled mechanisms (because ground truth is known) and stress-test capabilities (abstraction/abduction, robustness to noise/distractors, handling interactions) that also matter in real discovery but cannot be evaluated precisely without ground truth. So the synthetic-vs-real gap is not meant to be a “surprising claim about the world,” but a useful signal that current methods have clear limitations under controlled conditions, and that these controlled datasets provide a reliable way to measure progress over time.
> >
> > Relatedly, asking for “training on synthetic data improves real discovery” is beyond the scope of a benchmark paper and also not something prior benchmarks typically require. For example, DiscoveryBench also reports that synthetic settings are harder than real ones, and does not frame the benchmark’s value around training on synthetic to improve real. We will clarify this intended role of synthetic datasets more directly in the revision.
> >
> > **Re: Similarity to DiscoveryBench**
> >
> > We agree DiscoveryBench is closely related and we discussed it in the paper. However, we believe HypoBench makes a significant and complementary contribution that is important for TMLR readers:
> >
> > 1. **Abstraction / abductive feature discovery**: DiscoveryBench largely assumes relevant features are already identified and structured, while HypoBench benchmarks the harder setting where models start from more unstructured observations and must propose latent variables themselves, and this is present in both our real and synthetic tasks.
> > 2. **Evaluation scope**: DiscoveryBench centers on ground-truth discovery, while HypoBench emphasizes explanatory power via practical utility and generalizability (IND/OOD and cross-model evaluation), and then adds preliminary interestingness metrics.
> > 3. **Comprehensive experiments**: DiscoveryBench primarily evaluates a smaller set of agentic frameworks/models, while HypoBench benchmarks combinations of multiple backbone models and multiple hypothesis generation methods under a unified protocol.
> >
> > We agree we can make these differences easier to see in the writing, and we will strengthen the comparison section accordingly.
> >
> > **Re: Requested changes / pitch**
> >
> > This is fair feedback. We agree the current writing may read as too centered on synthetic design. In the revision, we will:
> > - Expand and clarify the motivation and novelty of both our real and synthetic tasks and their abstraction layer
> > - Reframe synthetic datasets as enabling ground-truth evaluation and controlled diagnosis
> >
> > Overall, we want to emphasize that HypoBench is a benchmark paper that provides (i) new datasets spanning real and controlled settings, (ii) evaluation protocols centered on explanatory power and generalization, and (iii) a broad comparison across models and hypothesis-generation methods. We strongly believe these contributions are valuable to the TMLR audience, and we will revise the paper to communicate them more clearly.

---

> > > ### Comment · Action_Editor_d1Fd · 2025-12-27
> > >
> > > Dear Reviewer aLXP,
> > >
> > > The authors have responded to your questions/concerns regarding the hypothesis discovery rate and the connection to the explanatory power focus, the authors' choice of models, the relationship to DiscoveryBench, among others. Please be sure to read through their response.
> > >
> > > Best,\
> > > AE

---

> ### Comment · Reviewer_aLXP · 2026-01-16
> **Response to the authors**
>
> I still have two fundamental issues with the paper:
>
> 1. The desideratum that one should measure the explanatory power of the hypothesis is great. However, its manifestation in the benchmark is really narrow. It just feels like the contrived choice of decision trees and logistic regression as function are made so that it fits with the traditional ML/curve fitting paradigm, but data-driven discovery goes beyond that. Most importantly, there is a distinguished literature on symbolic regression where there are several benchmarks, on real data, that study the equation (as a form of hypothesis) discovery through their test-time predictive power.
>
> Some refs:
> 1) LLM-SR: Scientific Equation Discovery via Programming with Large Language Models
> 2) LLM-SRBench: A New Benchmark for Scientific Equation Discovery with Large Language Models
>
> So, in that regard, I do not really see the contributions of this benchmark standing out.
>
> 2. The fact that the main contribution revolves around showing that models/agents do not perform well on their synthetic setup. Even if they could expand the choice of synthetic functions (and even with noise and distractors), it does not answer anything about these models'/agents' performance on real data. In fact, the paper fails to give a single clear example where the low performance of current methods would transfer in a real setting where it will be detrimental. In fact, that's what past works tried to do: painfully collecting real datasets, and still tried to evaluate methods' performance that directly speaks to their performance in a real setting.
>
> Unfortunately, with the current state of the paper, content in the rebuttal, and the amount of revision that might be needed for the paper to get published in TMLR, I recommend "reject".

---

> ### Author Response · Authors · 2026-01-16
>
> Thank you for your continued engagement. We appreciate the additional references to symbolic regression benchmarks. However, we respectfully but firmly disagree with both points raised, and we believe there may be a persistent misunderstanding about our benchmark's design and contributions.
>
> **Re: Symbolic regression comparison and "contrived" function choices**
>
> We want to be clear: HypoBench is fundamentally different from symbolic regression benchmarks like LLM-SR and LLM-SRBench. In symbolic regression, the task is to discover a mathematical equation given structured, pre-defined variables (e.g., given measurements of mass, velocity, and energy, find E = mv²). The variables are explicitly provided, and the challenge is to discover the underlying function.
>
> For a large portion of datasets in HypoBench, **models do not have access to pre-defined variables**. Instead, they receive unstructured natural-language observations and must:
> 1. Identify latent variables through abductive reasoning (e.g., inferring that "confidence" or "emotional tone" might be relevant from raw text descriptions)
> 2. Propose relationships between these inferred variables and outcomes
>
> This abstraction layer, which we have emphasized repeatedly (page 5, and in our previous responses), makes HypoBench a test of the full hypothesis generation pipeline, not just curve fitting on known features. Please also check the examples of such pipelines in table 3, page 6. The choice of decision trees and logistic regression as underlying ground-truth mechanisms is intentional, as they are interpretable and allow precise evaluation. The task presented to models is not "fit this function to these variables." The task is "read these observations and propose what hidden factors might explain the outcomes."
>
> We acknowledge that symbolic regression is related work, and we are happy to add additional discussions in Related Work. However, we do not believe HypoBench should be evaluated as if it were a symbolic regression benchmark, because the core challenges are different.
>
> **Re: "Main contribution revolves around showing models do not perform well on synthetic setup"**
>
> We respectfully disagree with this claim. Our benchmark’s main contributions are:
>
> 1. We curate 194 datasets for hypothesis generation, spanning 7 real-world and 5 synthetic domains, with real-world tasks covering deceptive reviews, AI-generated content detection, persuasion, mental health, and more.
>
> 2. We propose a principled evaluation framework measuring explanatory power through practical utility (classification accuracy on held-out data), generalizability (IND/OOD evaluation), and HDR for ground-truth recovery on synthetic tasks.
>
> 3. We perform systematic evaluations of 4 LLMs × 6 hypothesis generation methods, highlighting that Qwen and Literature+Data performs best in our models and methods selections. Additionally, we found that models struggle in (1) generating hypotheses with complex interactions involving more than 2 hidden variables and (2) uncovering counterintuitive hypotheses, suggesting that prior knowledge can hurt discovery if the ground-truth contradicts expectations.
>
> These contributions are not simply "models fail on synthetic data." In addition to the dataset and evaluation contributions, the findings from the synthetic datasets are specific, actionable findings about **where and why current models and methods have limitations**, and such findings cannot be obtained without ground-truth controlled settings.
>
> We also note that the framing that “synthetic benchmarks must demonstrate transfer to real settings to be valuable” is not a standard expectation for benchmarks. DiscoveryBench similarly uses synthetic tasks for ground-truth evaluation without requiring such transfer demonstrations. The purpose of synthetic benchmarks is controlled diagnosis and complementing (not replacing) real-world evaluation. HypoBench provides both.
>
> We have worked to address all feedback constructively across two revision rounds, including a human annotation study validating our metrics and an O3 evaluation demonstrating continued benchmark difficulty. We believe that HypoBench provides meaningful value to the community as a comprehensive benchmark for hypothesis generation.

---

> ### Author Response · Authors · 2026-01-16
> **Hypothesis generation pipeline example**
>
> We provide examples of our full hypothesis generation pipelines below:
>
> **Task: Retweet (Real)**
> ```
> Input (x): First tweet: "CNN: Senate Democrats supported rule that led to insurance cancellations."
> Second tweet: "Senate Dems knew millions would receive cancellation notices, because they voted for it."
>
> Label (y): 1 (second tweet got more retweets)
>
> Ground-truth features (Z): N/A
>
> Ground-Truth predictor (f): N/A
>
> Generated Hypothesis:
> "Tweets that engage the audience by addressing them directly or using inclusive language tend to receive more retweets than those that focus solely on the author's perspective."
> ```
>
> **Task: Presidential Election (Synthetic)**
> ```
> Input (x):
> "I gave up on the two party thing a while ago. But that Supreme Court decision still bothered me. It felt very one sided, like they always favor conservatives. Meanwhile we're still debating whether corporations need more tax breaks?? I care more
>   about helping people in my community. That's where I put my energy now."
>
> Label (y): likely third-party/abstain voter
>
> Ground Truth Features (Z):
> {
>   # Categorical variables
>   political_endorsement: "criticizes mainstream political parties",
>   policy_stance: "opposes tax cuts for corporations",
>   partisan_language: "supports social justice initiatives",
>   political_event_reaction: "criticizes Supreme Court decision favoring conservatives"
> }
> ```
>
> **Ground Truth Predictor (f):**
> A multinomial logistic regression model computes the predicted label as: $f(Z) = \arg\max_k \left( W_k \cdot Z \right)$, where $Z \in \{0,1\}^d$ is the binary feature vector indicating the presence of each feature, and $W \in \mathbb{R}^{K \times d}$ is the class weight matrix. In this example, features like *opposes tax cuts* and *supports social justice* contribute strongly to the *Democratic* class score, outweighing any ambiguity introduced by *criticizes mainstream political parties*.
>
> ```
> Ground Truth Hypothesis (f(Z) in natural language):
> Individuals who criticize conservative-leaning institutions, advocate for social justice, and oppose corporate tax cuts are likely to support the Democratic candidate, even when they express dissatisfaction with the broader political system.
>
> Generated Hypothesis:
> Tweets expressing support for Democratic policies like universal healthcare and climate action tend to indicate a Democratic voter.
> ```

---

### Author Response · Authors · 2025-12-29
**Revision summary and followups**

We thank all three reviewers for their thoughtful feedback. We have submitted a revision with the recommended changes:
- **Added Limitations section** (Section 8) discussing evaluation methodology, scope, and synthetic vs. real-world interpretation.
- **Restructured Evaluations section** to emphasize practical utility as the primary metric, with HDR as a complementary diagnostic for synthetic datasets.
- **Added "Abstraction layer" paragraph** on page 4, clarifying that our synthetic setup tests the full pipeline from unstructured text to hypotheses, which differs from previous works such as DiscoveryBench.
- **Added motivation for logistic regression/decision trees** (page 4) as interpretable, complementary models enabling precise ground-truth evaluation.
- **Added motivation of synthetic datasets** on page 10, explaining they diagnose specific capabilities (noise sensitivity, feature interactions, distractors) that cannot be measured without ground truth.
- **Added discussion of D5** (Zhong et al.) in Related Works, clarifying how our work differs from comparative corpus analysis.
- **Clarified zero-shot inference vs. zero-shot generation** on page 8.
- **Changed table formatting** to 1 decimal place for percentages.

To further address the concerns about LLM-as-a-judge approach in our HDR metric, we will follow-up with a small experiment comparing agreements between human annotated HDR scores with LLM-as-a-judge scores.

We are also running a set of follow-up experiments for answering the question from reviewer qmYN, *Can the latest reasoning models solve the datasets already*.

We will upload another revision with the two experiments above early next week.

---

> ### Author Response · Authors · 2026-01-05
> **Additional revision**
>
> Happy New Year to all reviewers and AE!
>
> We have uploaded another revision with the following changes:
>
> - **Human Annotation Study** (Section 5, end of Results): Validated LLM-as-judge metrics against human judgment on 100 hypothesis pairs. Human-human agreement was substantial to almost perfect ($κ=0.80$ for FDR, $κ_w=0.86$ for RC). Model-human agreement was substantial ($κ=0.71$ for FDR, $κ_w=0.64$ for RC), confirming that LLM evaluation provides reliable approximations of human judgment. Updated Limitations section to reference this validation.
>
> - **O3 Evaluation** (Appendix C.4): Tested whether advanced reasoning models can "solve" HypoBench. O3 achieved average HDR of 0.52 across 11 synthetic datasets, on average with good performance on feature discovery (0.99 FDR) and moderate relationship correctness (0.52 RC). Specifically, O3 can solve the easier datasets from HypoBench but fails the higher difficulty ones (0.25 HDR). Compared to HypoGeniC with GPT-4o-mini, O3 outperformed on some tasks (admission/level_3: 0.45 HDR vs 0.30 HDR) but underperformed on others (shoe_simple: 0.75 HDR v.s. 0.88 HDR). which validates the benchmark's difficulty. We also referred to this in our limitation section.
>
> We would like to thank again for the AE and all reviewers' help. We are dedicated to improve HypoBench and make it a useful resource for the community.

---

### Author Response · Authors · 2026-01-06
**Response and revision summary**

Dear AE,

Thank you for your continued guidance throughout this review process, and Happy New Year to you and all reviewers! We appreciate the thoughtful feedback from all three reviewers, which has helped us strengthen the paper. We are also pleased that Reviewer RaBX found the claims in the submission "supported by accurate, convincing and clear evidence" and recognized the relevance of hypothesis generation to the TMLR audience.

**Addressing Reviewer Concerns:**

We have made two rounds of revisions addressing all major concerns:

*Reviewer aLXP* raised concerns about HDR's role, synthetic data motivation, and similarity to DiscoveryBench.
- We restructured the Evaluations section to emphasize practical utility as the primary metric, with HDR as a complementary evaluation (Section 5) when ground-truth hypotheses are available.
- We added explicit motivation for synthetic datasets, explaining they enable ground-truth evaluation of capabilities that cannot be measured otherwise.
- We added an "Abstraction layer" paragraph clarifying that our setup tests the full pipeline from unstructured text to hypotheses, which is a key distinction from DiscoveryBench.

*Reviewer RaBX* requested clarifications on metrics, diversity, and formatting.
- We changed table formatting to 1 decimal place for readability.
- We added a capability-by-task mapping and addressed questions about statistical significance in our response.

*Reviewer qmYN* raised concerns about LLM-as-judge validity and whether reasoning models can solve the benchmark.
- **Human Annotation Study (Section 5):** We validated LLM-as-judge against human judgment on 100 hypothesis pairs, finding substantial agreement between model and human scores, confirming that LLM evaluation reliably approximates human judgment.
- **O3 Evaluation (Appendix C.4):** We tested OpenAI's O3 reasoning model, which achieved moderate overall performance but struggled on higher-difficulty tasks, validating the benchmark's continued challenge.
- We added discussion of D5 (Zhong et al.) in Related Works.

**Additional Changes:**
- Added Limitations section (Section 8)
- Clarified zero-shot inference vs. zero-shot generation (Table 5)
- Added motivation for logistic regression/decision trees as interpretable models

**Summary of Contributions:**

To reiterate, our main contributions are:

1. We developed HypoBench, a comprehensive benchmark for evaluating hypothesis generation with 194 datasets spanning 12 domains (7 real-world, 5 synthetic). Our synthetic datasets provide ground-truth hypotheses for precise capability diagnosis.

2. We proposed a principled evaluation framework emphasizing explanatory power through practical utility and generalizability (IND/OOD), complemented by HDR for ground-truth recovery on synthetic tasks.

3. We conducted systematic evaluations of 4 state-of-the-art LLMs and 6 hypothesis generation methods, demonstrating significant room for improvement and the value of integrating literature with data.

We believe these revisions comprehensively address reviewer concerns and strengthen the paper. HypoBench fills an important gap by providing the first systematic benchmark for hypothesis generation with both real-world and controlled synthetic evaluation. We are confident this work will be a valuable resource for the TMLR community.

Thank you again for your time and consideration. Please let us know if any further clarification is needed.

Best regards,
HypoBench Authors

---

### Decision · Action_Editor_d1Fd · 2026-02-23

**Recommendation:** Reject

**Additional Comments:**

It is evident from the discussion and the updates to the paper that the authors made a concerted effort to address the reviewers' questions and concerns. Several issues initially raised by the reviewers were resolved, including the role of the hypothesis discovery rate (HDR) metric as being akin to a recall measure; the choice of logistic regression and decision tree models due to their interpretability; the difference between HypoBench and symbolic regression; the performance of reasoning models on the benchmark; as well as several other points of clarification.

Much of the discussion between the reviewers, authors, and AE focused on the role of the synthetic components of the benchmark and the extent to which the conclusions drawn from these controlled experiments inform the hypothesis generation capabilities of LLMs in real-world settings. Reviewers, particularly Reviewer aLXP, remain concerned that the performance gaps observed on the synthetic datasets do not have clear implications for a model's performance in real-world domains. The AE appreciates the fact that the synthetic domains provide a controlled means to identify a model's different failure modes, as the authors emphasize; however, there are still questions about the practical relevance of these failure modes, i.e., beyond synthetic domains.

Related, Reviewer qmYN raised concerns about the reliance on LLM-based judges (GPT-4o) for computing several of the evaluation metrics (e.g., novelty, plausibility, clarity, and HDR) and, in turn, about the broader significance of the associated results. In response, the authors added a comparison between human judgments and LLM-as-judge scores that shows meaningful human-model agreement. The results are encouraging, but the experiments involved only two human annotators and were limited to HDR, leaving out the other metrics that require LLM-based judges, including those used to evaluate performance on the real-world datasets (namely, novelty, plausibility, and clarity).

After the rebuttal period, one of the reviewers leans towards acceptance, another leans towards rejection, and the third recommends rejection. The AE discussed the paper with the reviewers who were most critical to better understand their opinions of the paper considering the authors' extensive efforts to respond to the initial reviews. This included a discussion of the authors' concerns about the initial reviews and the rebuttal phase, notably contributions relative to existing symbolic regression work, one reviewer's suggestion that the contributions beyond the synthetic datasets and results are limited, questions about HypoBench's value relative to existing benchmarks, and the impact of reliance on LLM-based judges in light of the recently added human judge comparison.

The AE believes that HypoBench is promising as a means to benchmark LLM-based hypothesis generation capabilities in practice, particularly in its focus on hypothesis generation from unstructured inputs via abductive reasoning. The paper could realize this potential by providing a stronger and more thorough validation of the LLM-as-judge evaluation metrics and by including a discussion/demonstration of how results on the synthetic domains provide actionable or generalizable insights into real-world hypothesis generation.

**Audience:**

No

**Audience Explanation:**

The reviewers offered mixed views on whether the findings would be of interest to the TMLR community. The AE agrees with the reviewers that the hypothesis generation capabilities of LLMs are, broadly speaking, an important and timely topic. However, the reviewers questioned whether the insights that can be gained from the HypoBench benchmark and the analyses, in their current form, are sufficiently developed to be of interest to the community. Underlying this concern are questions about the practical implications of the synthetic dataset results, due to the reliance on LLM-based judges and the perceived lack of evidence that performance on synthetic domains provides meaningful insights into a model's behavior on real-world problems. The AE recognizes that the paper's contributions extend beyond the synthetic datasets and results, including an analysis of practical utility as a measure of explanatory power and generalizability, and agrees that it is incorrect, as at least one reviewer suggests, to reduce the paper to providing only synthetic findings. Given that the synthetic datasets and results are conveyed as a significant component of HypoBench, however, the AE recognizes the need to more clearly convey their relevance to model performance beyond the particular domains themselves.

**Claims And Evidence:**

Yes

**Claims Explanation:**

The paper proposes HypoBench, a benchmark to evaluate the hypothesis generation capabilities of large language models (LLMs). HypoBench utilizes two metrics, practical utility and the hypothesis discovery rate (HDR), as a means of measuring the explanatory power of the generated hypotheses. Additionally, the paper proposes additional metrics to evaluate the "interestingness" of the resulting hypotheses. The benchmark provides 194 datasets for evaluating hypothesis generation based on seven real-world domains and five synthetic domains. The paper utilizes the benchmark to evaluate the capabilities of four contemporary LLMs using six hypothesis generation methods.

The claims made in the paper primarily involve the inclusion of 194 curated datasets from real-world and synthetic domains, the proposal of metrics for evaluating the explanatory power and "interestingness" of LLM-based hypothesis generation, and the systematic evaluation of four different LLMs and six hypothesis generation methods. The paper supports these claims.

**Resubmission Of Major Revision:**

The authors may consider submitting a major revision at a later time.